

# Dipole symmetries from the topology of the phase space and the constraints on the low-energy spectrum

Tomáš Brauner[1⋆], Naoki Yamamoto[2] and Ryo Yokokura[3]

**1** Department of Mathematics and Physics, University of Stavanger, 4021 Stavanger, Norway
**2** Department of Physics, Keio University, Yokohama 223-8522, Japan
**3** Department of Physics & Research and Education Center for Natural Sciences, Keio University, Yokohama 223-8521, Japan

⋆ tomas.brauner@uis.no

## Abstract

We demonstrate the general existence of a local dipole conservation law in bosonic field theory. The scalar charge density arises from the symplectic form of the system, whereas the tensor current descends from its stress tensor. The algebra of spatial translations becomes centrally extended in presence of field configurations with a finite nonzero charge. Furthermore, when the symplectic form is closed but not exact, the system may, surprisingly, lack a well-defined momentum density. This leads to a theorem for the presence of additional light modes in the system whenever the short-distance physics is governed by a translationally invariant local field theory. We also illustrate this mechanism for axion electrodynamics as an example of a system with Nambu–Goldstone modes of higher-form symmetries.

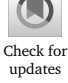

# 1 Introduction and summary

Multipole symmetries have played an important role for understanding the restricted mobility of elementary excitations in fracton phases of matter [1–4]. The simplest example, a dipole-type symmetry, amounts to a local conservation law

$$\partial_0 J^0 + \partial_i \partial_j J^{ij} = 0 \,, \tag{1}$$

where $J^0$ is a charge density and $J^{ij}$ a symmetric spatial tensor current. On field configurations such that $J^{ij}$ decays sufficiently rapidly at spatial infinity, (1) implies the conservation of the following quantities,[1]

$$Q \equiv \int \mathrm{d}^d\boldsymbol{x}\, J^0(\boldsymbol{x}, t)\,, \qquad D^i \equiv \int \mathrm{d}^d\boldsymbol{x}\, x^i J^0(\boldsymbol{x}, t)\,. \tag{2}$$

If in addition the spatial tensor current is traceless, $\delta_{ij} J^{ij} = 0$, then

$$X \equiv \int \mathrm{d}^d\boldsymbol{x}\, \boldsymbol{x}^2 J^0(\boldsymbol{x}, t)\,, \tag{3}$$

is also conserved. Clearly, $D^i$ and $X$ can be interpreted respectively as the dipole moment and trace of the quadrupole moment of the charge density $J^0$. The most striking consequence of dipole-type conservation laws is the restriction on motion of any localized field configuration with finite nonzero $Q$. Namely, for such excitations, $D^i/Q$ plays the role of a center of charge. Conservation of both $Q$ and $D^i$ implies that the mean position of the excitation cannot change, even though its detailed spatial profile may vary with time. It has been known that apart from as yet hypothetical fracton phases of matter, a dipole-type conservation law also restricts the mobility of skyrmions in ferromagnets [5] and vortices in superfluids [6]. A similar mobility constraint also appears in relativistic physics when so-called Carrollian symmetry is present [7].

---

[1] Unless explicitly stated otherwise, we will have implicitly in mind physical systems living in a $d$-dimensional Euclidean space, $\mathbb{R}^d$, throughout the paper.

Physical systems featuring a given multipole symmetry can be constructed using standard symmetry-based techniques of effective field theory (EFT) [8, 9]. However, the question in what kind of systems one may actually expect a multipole symmetry does not seem to have a clear answer. In this paper, we show that a dipole-type conservation law appears in nearly any local bosonic field theory with continuous translation invariance; the only mild technical assumption we make is that the Lagrangian of the system does not contain higher than first time derivatives or mixed spatial-temporal derivatives. This is the subject of Sec. 2. The main takeaway point is that $J^0$ is the density of a topological charge, associated with the symplectic form of the system. The tensor current $J^{ij}$ descends from the system's stress tensor.

The origin of the dipole conservation law in the symplectic form of the system suggests that one may gain further insight by switching to the Hamiltonian (symplectic) formalism. Restricting the discussion temporarily to $d = 2$ spatial dimensions, we thus show in Sec. 3 that $D^i$ and $X$ are essentially the generators of (two-dimensional) spatial translations and rotations. The precise relation to the operators of momentum $P_i$ and angular momentum $L$ is

$$P_i = -\epsilon_{ij}D^j, \qquad L = \frac{1}{2}X.$$ (4)

Moreover, the algebra of spatial momentum is centrally extended,

$$\boxed{\{P_i, P_j\} = -\epsilon_{ij}Q.}$$ (5)

This kind of central extension was previously shown to be present in EFTs for Nambu–Goldstone (NG) bosons of spontaneously broken internal symmetry, and in EFTs with fields taking values from a Kähler manifold [10]. The present derivation generalizes this result to all bosonic field theories satisfying our mild technical assumption on the dependence of the Lagrangian on time derivatives of the fields.

The above general results are illustrated by examples in Sec. 4. We revisit skyrmions in ferromagnets and vortices in superfluids, pointing out the essential difference in how topological textures and defects fit into the framework developed in this paper. We also discuss a dipole-type conservation law in quantum Wigner crystals.

In $d \geq 2$ spatial dimensions, the symplectic 2-form generates a $(d-2)$-form symmetry (see [11–13] for recent reviews of generalized symmetries). For $d > 2$, one therefore cannot expect the corresponding topological charge to appear in the Poisson bracket (or commutator) of momentum components as in (5). This is in accord with the fact that in rotationally invariant systems in $d > 2$ dimensions, any central extension of the Euclidean algebra of momentum and angular momentum is forbidden by the Jacobi identities. A proper generalization of (5) is presented in Sec. 5.

The presence of a central extension in the momentum algebra is not innocuous. Namely, it may indicate that one cannot define a consistent momentum density, a feature sometimes referred to as the *linear momentum problem* (LMP). In Sec. 6 we interpret this as a "classical anomaly" [14] and show how it restricts the low-energy spectrum of the system. The constraint is particularly strong for gapless modes, that is NG bosons. Here it ultimately leads to a no-go theorem for certain types of symmetry-breaking patterns unless additional gapless degrees of freedom are present; see Sec. 7 for details. In Sec. 8 we show on a concrete example that the prediction of additional gapless modes also applies to NG bosons of spontaneously broken higher-form symmetry.

Some additional comments on the material presented in the paper are offered in Sec. 9. Several technical details that we omit in the main text are relegated to appendices. In Appendix A, we show explicitly how the LMP is avoided in ferromagnetic insulators. Appendix B demonstrates the absence of a well-defined momentum density in axion electrodynamics. Appendices C and D further extend the general results of this paper. First, it is common to express

local conservation laws in a coordinate-free form as the statement of closedness of the Hodge dual of the symmetry current. It is not immediately obvious how the dipole conservation law (1) could be cast in such a coordinate-free form as well. This is the subject of Appendix C, where we also mention possible applications to systems living on curved spatial manifolds. Finally, the consequences of global symmetry are oftentimes conveniently expressed in terms of invariance of the generating functional of the theory under gauge transformations of a set of background fields. The background gauge invariance corresponding to the class of dipole-type symmetries considered here is worked out in Appendix D. This provides a natural link between the present paper and the recent work of Du et al. [15] on invariance of systems with fracton-like mobility constraints under volume-preserving diffeomorphisms.

## 2 Dipole conservation laws from translation invariance

To motivate our general framework, consider a theory of a set of bosonic (but not necessarily scalar) fields $\chi^a$ taking values from a manifold $\mathcal{N}$, defined by the Lagrangian density $\mathscr{L}[\chi]$. We assume that the Lagrangian does not contain any higher than first time derivatives, or mixed spatial-temporal derivatives, of $\chi^a$. The set of corresponding conjugate momenta $\Pi_a$ is then defined in the usual way, $\Pi_a \equiv \partial\mathscr{L}/\partial(\partial_0\chi^a)$. Under our assumption on the Lagrangian, this is a local algebraic relation between the generalized velocities $\partial_0\chi^a$ and the conjugate momenta $\Pi_a$. We assume furthermore the absence of constraints so that this relation can be inverted to uniquely give $\partial_0\chi^a$ as a function of $\Pi_a$, $\chi^a$ and the spatial derivatives of $\chi^a$. The Hamiltonian density $\mathscr{H} \equiv \Pi_a\partial_0\chi^a - \mathscr{L}$ can then be likewise expressed in terms of $\Pi_a$ and $\chi^a$ and their spatial derivatives. Locally, one may view the pairing $\Pi_a\partial_0\chi^a$ as defining the action of a 1-form on tangent vectors to $\mathcal{N}$. The phase space of the theory then consists of maps from $\mathbb{R}^d$ to $T^*\mathcal{N}$, the cotangent bundle of $\mathcal{N}$.

It is not always convenient to explicitly separate generalized coordinates from their conjugate momenta. Moreover, there are systems where the Lagrangian dynamics is naturally first-order, in which case the set of variables $\chi^a, \Pi_a$ cannot be treated as independent, or unconstrained. In order to allow for such possibility, we will now formulate the class of theories we shall be concerned with directly in terms of a Hamiltonian action principle. We label all the independent canonical variables jointly as $\phi^a$ and assume that they take values from a manifold $\mathcal{M}$. To distinguish the finite-dimensional manifold $\mathcal{M}$ from the actual phase space, which is the infinite-dimensional set of maps $\phi^a : \mathbb{R}^d \to \mathcal{M}$, we will refer to $\mathcal{M}$ as the *target space* of the theory. The action of the theory is then defined as

$$S = \int \mathrm{d}^d\boldsymbol{x}\,\mathrm{d}t\,\{\omega_a(\phi)\partial_0\phi^a - \mathscr{H}[\phi]\}\,; \tag{6}$$

the Hamiltonian density $\mathscr{H}[\phi]$ is a local function of the fields $\phi^a$ and their spatial derivatives. The 1-form $\omega(\phi) \equiv \omega_a(\phi)\mathrm{d}\phi^a$ on $\mathcal{M}$ defines the symplectic potential of the theory. The absence of further constraints is embodied in the assumption that the symplectic 2-form, $\Omega \equiv \mathrm{d}\omega$, is nondegenerate on the entire target space $\mathcal{M}$.[2]

Suppose that the theory possesses global translation invariance, that is, neither the Hamiltonian nor the symplectic potential depends explicitly on the spacetime coordinates $x^\mu = (\boldsymbol{x}, t)$. Let us now subject the action to an infinitesimal spatial, possibly time-dependent, diffeomorphism, $x^i \to x^i + \xi^i(\boldsymbol{x}, t)$. Then $\omega_\mu \equiv \omega_a\partial_\mu\phi^a$ transforms as a spacetime 1-form,

---

[2]This is a slight abuse of terminology. Usually, the symplectic potential and symplectic form are defined directly as differential forms on the phase space rather than on the target space $\mathcal{M}$. We will however only work with differential forms on $\mathcal{M}$ and their pull-back to the space(time), so there is no danger of confusion.

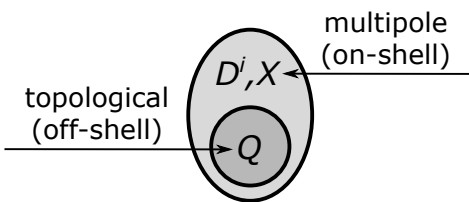

Figure 1: Nested conservation laws. The topological charge $Q$ is conserved off-shell. Imposing the equation of motion leads to two additional conserved quantities: the dipole moment $D^i$ and the trace of the quadrupole moment $X$.

whereas the variation of the Hamiltonian, $H = \int d^d \mathbf{x} \, \mathscr{H}$, defines the symmetric stress tensor $\sigma_{ij}$,

$$\delta_\xi \omega_\mu = -\xi^i \partial_i \omega_\mu - \omega_i \partial_\mu \xi^i, \qquad \delta_\xi H \equiv \int d^d \mathbf{x} \, \frac{1}{2}(\partial^i \xi^j + \partial^j \xi^i)\sigma_{ij} = \int d^d \mathbf{x} \, \partial_i \xi^j \sigma^i{}_j. \tag{7}$$

Inserting this into the action and integrating by parts, we get

$$\delta_\xi S = \int d^d \mathbf{x} \, dt \, \xi^i (\partial_0 \omega_i - \partial_i \omega_0 + \partial_j \sigma^j{}_i). \tag{8}$$

For fields that satisfy the equation of motion (that is, are *on-shell*), this implies the relation

$$\partial_0 \omega_i - \partial_i \omega_0 = -\partial_j \sigma^j{}_i. \tag{9}$$

For the sake of simplicity, let us next initially assume $d = 2$ spatial dimensions. The symplectic potential can then be used to construct the current

$$J^\mu \equiv \epsilon^{\mu\nu\lambda} \partial_\nu \omega_\lambda. \tag{10}$$

This gives a topological conservation law in the sense that the current is conserved identically (*off-shell*). The corresponding charge density essentially equals the pull-back of the symplectic 2-form on $\mathcal{M}$, $\Omega = d\omega$, to $\mathbb{R}^2$,

$$\boxed{J^0 = \epsilon^{ij} \partial_i \omega_j.} \tag{11}$$

On-shell, the spatial part of the topological current can be related to the stress tensor via (9),

$$J^i = \epsilon^{ij}(\partial_j \omega_0 - \partial_0 \omega_j) = \partial_j(\epsilon^{ik} \sigma^j{}_k). \tag{12}$$

Upon introducing the symmetric tensor current,

$$\boxed{J^{ij} \equiv \frac{1}{2}\big(\epsilon^{ik} \sigma^j{}_k + \epsilon^{jk} \sigma^i{}_k\big),} \tag{13}$$

the conservation of the topological current $J^\mu$ turns into the dipole conservation law (1).[3] The tensor current $J^{ij}$ is traceless as a consequence of the symmetry of $\sigma^{ij}$.

Let us stress that what we have here are two different conservation laws that are *nested*; see Fig. 1 for a sketch. The current $J^\mu$, and hence the integral charge $Q$, is conserved off-shell.

---

[3]The same identification of $J^0$ with a topological charge density and $J^{ij}$ with the stress tensor appeared previously in the specific contexts of ferromagnetism [5] and chiral topological elasticity [16].

If we now on top of that impose the equation of motion, we get the on-shell relation (12). This guarantees via (1) the conservation of $D^i$ and $X$ as defined by (2) and (3).

Translation invariance is usually associated with conservation of momentum. In order for (1) to have any additional content, there must be field configurations for which $Q$ actually is nonzero. There are two generic options how to ensure this. First, it is often useful to impose a boundary condition on the fields that effectively compactifies the domain $\mathbb{R}^2$ to a manifold without boundary such as the sphere $S^2$ or torus $T^2$. The image of the fields $\phi^a$ thus becomes a 2-cycle in the target space $\mathcal{M}$. The integral charge $Q$ can then be nonzero only if this 2-cycle belongs to a nontrivial homology class in $\mathcal{M}$, and at the same time the symplectic form $\Omega = d\omega$ is closed but not exact.[4] The second option is that the fields $\phi^a$ satisfy a nontrivial boundary condition, whereby they map $\mathbb{R}^2$ to a 2-chain in $\mathcal{M}$ with a nonvanishing boundary. The topological charge $Q$ may then be nonzero even if the symplectic form $\Omega$ is exact, as we will see on a concrete example in Sec. 4.

While we have initially restricted the discussion to $d = 2$ spatial dimensions, the generalization to higher dimensions is easy. The on-shell relation (9) remains valid for any $d$. What changes is only the definition of the topological current descending from the symplectic form,

$$J^{\mu_1 \cdots \mu_{d-1}} \equiv \epsilon^{\mu_1 \cdots \mu_{d-1} \nu \lambda} \partial_\nu \omega_\lambda, \tag{14}$$

giving rise to a $(d-2)$-form topological conservation law. Following the same steps as above then leads to the generalized dipole-type conservation law,

$$\partial_0 \rho^{i_1 \cdots i_{d-2}} + \partial_j \partial_k J^{i_1 \cdots i_{d-2} jk} = 0. \tag{15}$$

Here the charge density $\rho^{i_1 \cdots i_{d-2}}$ and the spatial tensor current $J^{i_1 \cdots i_{d-2} jk}$, antisymmetric in its first $d-2$ indices and symmetric in the last two indices, are defined by

$$\rho^{i_1 \cdots i_{d-2} jk} \equiv \epsilon^{i_1 \cdots i_{d-2} jk} \partial_j \omega_k, \qquad J^{i_1 \cdots i_{d-2} jk} \equiv \frac{1}{2}\left(\epsilon^{i_1 \cdots i_{d-2} j\ell} \sigma^k{}_\ell + \epsilon^{i_1 \cdots i_{d-2} k\ell} \sigma^j{}_\ell\right). \tag{16}$$

In other words, the conservation law (1) augmented with the definitions (11) and (13) is modified trivially by adding an antisymmetrized set of extra spatial indices $i_1, \ldots, i_{d-2}$ on the currents and the Levi-Civita tensor. We stress that the result (15) is still a dipole-type conservation law; it is not to be confused with a multipole-type conservation law, which would be symmetric in all its indices.

## 3 Symplectic approach to the dipole algebra in $d = 2$ dimensions

The local conservation law (1) was easy to generalize to any number of dimensions. However, the corresponding global conservation laws turn out to be qualitatively different for $d = 2$ and $d > 2$. We will therefore for the time being again restrict to $d = 2$ dimensions, and return to the general case in Sec. 5.

In this section, we will study the global symmetry algebra using the Poisson bracket, defined for any two functionals $F, G$ on the phase space by

$$\{F, G\} \equiv \int d^d x \, \Omega^{ab}(\phi(x)) \frac{\delta F}{\delta \phi^a(x)} \frac{\delta G}{\delta \phi^b(x)}. \tag{17}$$

Here $\Omega^{ab}$ is the matrix inverse of $\Omega_{ab}$, collecting the matrix elements of the symplectic form, defined by the usual prescription $\Omega \equiv d\omega \equiv (1/2)\Omega_{ab} d\phi^a \wedge d\phi^b$ where $\Omega_{ab} = \partial_a \omega_b - \partial_b \omega_a$,

---

[4]The latter condition is satisfied in particular whenever $\mathcal{M}$ is a compact manifold without boundary.

with the shorthand notation $\partial_a \equiv \partial/\partial\phi^a$. The key step is to introduce a deformation of the topological charge $Q$ in (2) with the charge density defined by (11),

$$Q_\lambda \equiv \int d^2x\, \lambda(x)\epsilon^{ij}\partial_i\omega_j(\phi(x)) = \frac{1}{2}\int d^2x\, \lambda(x)\epsilon^{ij}\Omega_{ab}(\phi(x))\partial_i\phi^a(x)\partial_j\phi^b(x), \qquad (18)$$

where $\lambda(x)$ is an arbitrary (smooth) function on $\mathbb{R}^2$. The topological nature of the charge $Q = Q_{\lambda=1}$ is reflected in the fact that all its Poisson brackets vanish. That is however no longer the case for $Q_\lambda$, whose Lie algebra reproduces that of functions on $\mathbb{R}^2$,

$$\{Q_\lambda, Q_{\bar\lambda}\} = -Q_{\epsilon^{ij}\partial_i\lambda\partial_j\bar\lambda} \equiv -Q_{\{\lambda,\bar\lambda\}}, \qquad (19)$$

where we have used that the symplectic form is closed, $\partial_a\Omega_{bc} + \partial_b\Omega_{ca} + \partial_c\Omega_{ab} = 0$. This is the so-called classical $w_\infty$ algebra [17]. Its local version [18],

$$\boxed{\{J^0(x), J^0(y)\} = -\frac{1}{2}\epsilon^{ij}\partial_i^x\partial_j^y\left\{[J^0(x) + J^0(y)]\delta(x-y)\right\},} \qquad (20)$$

is the coordinate-space formulation of the long-wavelength limit of the Girvin–MacDonald–Platzman algebra [19].

The functionals $Q_\lambda$ generate a nontrivial flow on the phase space,

$$\delta_\lambda\phi^a(x) \equiv \{\phi^a(x), Q_\lambda\} = \Omega^{ab}(\phi(x))\frac{\delta Q_\lambda}{\delta\phi^b(x)} = -\epsilon^{ij}\partial_i\lambda(x)\partial_j\phi^a(x). \qquad (21)$$

Now setting $\lambda(x) \to -\epsilon_{ij}x^j$ gives $\delta_\lambda\phi^a(x) \to -\partial_i\phi^a(x)$. Likewise, $\lambda(x) \to (1/2)x^2$ leads to $\delta_\lambda\phi^a(x) \to -\epsilon^{ij}x_i\partial_j\phi^a(x)$. Recognizing these respectively as an infinitesimal spatial translation and rotation, we can identify the total momentum and angular momentum as

$$P_i = Q_{-\epsilon_{ij}x^j} = -\epsilon_{ij}D^j, \qquad L = Q_{(1/2)x^2} = \frac{1}{2}X. \qquad (22)$$

Upon substituting the appropriate functions for $\lambda(x)$ and $\bar\lambda(x)$ in (19), we find that

$$\{L, P_i\} = \epsilon_i{}^j P_j, \qquad \{P_i, P_j\} = -\epsilon_{ij}Q. \qquad (23)$$

While the first relation copies the known properties of Euclidean symmetry transformations, the second one is a surprise. In presence of field configurations with a nonzero value of the topological charge $Q$, the algebra of spatial translations is centrally extended.

Before we illustrate this result on several examples, let us append a few comments. First, suppose that our classical action (6) constitutes a leading-order approximation to a quantum theory. The central extension of the classical Lie algebra of spatial translations can then be promoted to a commutator of the corresponding quantum operators,

$$[P_i, P_j] = -i\epsilon_{ij}Q. \qquad (24)$$

This leads in turn to a projective representation of the group of finite spatial translations,

$$e^{iu\cdot P}e^{iv\cdot P} = e^{iv\cdot P}e^{iu\cdot P}e^{i(u\times v)Q}, \qquad (25)$$

where $u, v \in \mathbb{R}^2$ are arbitrary translation vectors. Another, more general way to express this projective realization of translations is through a path-ordered exponential of the momentum

operator, integrated around a closed curve $\Gamma$ in $\mathbb{R}^2$. The result measures the (oriented) area $S_\Gamma$ of the domain, bounded by the curve $\Gamma$,

$$\mathcal{P}\exp\left(\mathrm{i}\oint_\Gamma \mathrm{d}\boldsymbol{x}\cdot\boldsymbol{P}\right) = e^{-\mathrm{i}S_\Gamma Q}. \tag{26}$$

The existence of such a phase accumulated upon transporting a topological soliton around a closed curve was predicted previously for skyrmions in ferromagnets [20].

Second, the general multipole algebra includes independent generators of spatial translations and rotations, alongside the multipole moments of the scalar charge $Q$ [21]. Our Lie algebra, generated by $\{Q, P_i, L\}$, is formally a contraction of a multipole algebra where the operators of momentum $P_i$ and angular momentum $L$ are identified with the dipole moment $D^i$ and trace of quadrupole moment $X$ via (22).

Finally, note that we have not obtained the integral momentum $P_i$ via Noether's theorem, but rather from (18) as a particular moment of the topological charge density. At first sight, one might therefore suspect that the central extension in (23) is a consequence of a specific choice of the momentum operator. Indeed, the question how to properly define the momentum of a ferromagnetic soliton has a long history with contributions spanning three decades; see for instance [5, 22–26]. We therefore stress that all Poisson brackets of a functional on the phase space are uniquely determined by the flow it generates, cf. (21). Poisson brackets of momentum are thus fixed by its definition as a generator of spatial translations. In other words, any two realizations of momentum can differ at most by a functional whose Poisson brackets identically vanish, that is a topological invariant on the phase space.

## 4 Examples

Below, we outline three different examples that represent three qualitatively different realizations of the centrally extended momentum algebra (5). In the first two, the topological charge $Q$ is realized through a cohomologically nontrivial symplectic form and field configurations that compactify the Euclidean plane $\mathbb{R}^2$ respectively to the sphere $S^2$ and the torus $T^2$. In the last example, the topological charge appears in spite of the symplectic form being exact, through field configurations satisfying a nontrivial boundary condition.

### 4.1 Ferromagnets

In ferromagnets, the SU(2) spin symmetry is spontaneously broken down to a U(1) subgroup. Accordingly, the long-distance physics of ferromagnets is captured by an EFT with fields taking values from the target space $\mathcal{M} \simeq \mathrm{SU}(2)/\mathrm{U}(1) \simeq S^2$. The phase space of two-dimensional ferromagnets consists of maps $\boldsymbol{n}(\boldsymbol{x}) : \mathbb{R}^2 \to S^2$, where $\boldsymbol{n} = \{n^A\}_{A=1}^3$ is a unit vector satisfying $\boldsymbol{n}^2 = 1$.[5] The symplectic structure is fixed by the local angular momentum algebra,

$$\{n^A(\boldsymbol{x}), n^B(\boldsymbol{y})\} = \frac{1}{M}\epsilon^{AB}{}_C n^C(\boldsymbol{x})\delta(\boldsymbol{x}-\boldsymbol{y}), \tag{27}$$

---

[5]All the following expressions are valid regardless of the choice of local coordinates $\phi^a$ on $S^2$. One particularly useful choice of coordinates, stressing the complex structure of the sphere, is $(Z, \bar{Z})$, where $Z \in \mathbb{C}$ is a complex variable that maps to the unit vector $\boldsymbol{n}$ via $n^1 = \frac{Z+\bar{Z}}{1+|Z|^2}$, $n^2 = \frac{1}{\mathrm{i}}\frac{Z-\bar{Z}}{1+|Z|^2}$, and $n^3 = \frac{1-|Z|^2}{1+|Z|^2}$. In terms of $(Z, \bar{Z})$, the symplectic 2-form (28) reads $\Omega = -\frac{2\mathrm{i}M}{(1+|Z|^2)^2}\mathrm{d}Z \wedge \mathrm{d}\bar{Z}$. Accordingly, the Poisson bracket of any functionals $F, G$ on the phase space of the ferromagnet is $\{F, G\} = -\frac{\mathrm{i}}{2M}\int \mathrm{d}^2\boldsymbol{x}(1+|Z|^2)^2\left(\frac{\delta F}{\delta Z}\frac{\delta G}{\delta \bar{Z}} - \frac{\delta G}{\delta Z}\frac{\delta F}{\delta \bar{Z}}\right)$.

where $M$ is the spin density in the ferromagnetic ground state. The Poisson bracket (27) in turn determines the symplectic 2-form on $S^2$,

$$\Omega = -\frac{M}{2}\epsilon_{ABC}n^A \mathrm{d}n^B \wedge \mathrm{d}n^C\,. \tag{28}$$

From (18), we infer that the topological charge $Q$ is given by

$$Q = -\frac{M}{2}\int \mathrm{d}^2\boldsymbol{x}\,\epsilon^{ij}\boldsymbol{n}\cdot(\partial_i\boldsymbol{n}\times\partial_j\boldsymbol{n}) = -4\pi M w[\boldsymbol{n}]\,, \tag{29}$$

where $w[\boldsymbol{n}]\in\mathbb{Z}$ is the Brouwer degree of the map $\boldsymbol{n}$. It follows from (23) that momentum in ferromagnets satisfies the Poisson bracket

$$\{P_i, P_j\} = 4\pi\epsilon_{ij}M w[\boldsymbol{n}]\,. \tag{30}$$

What are the field configurations for which $w[\boldsymbol{n}]$ is nonzero? The target space $\mathcal{M}\simeq S^2$ is compact and possesses a unique generator of second de Rham cohomology. This is exactly the symplectic 2-form, which equals up to normalization the area form on $S^2$. As a consequence, nonzero values of $Q$ can be achieved for smooth fields $\boldsymbol{n}(\boldsymbol{x})$ that map $\mathbb{R}^2$ to a homologically nontrivial 2-cycle on $S^2$. These are maps that effectively compactify $\mathbb{R}^2$ to a sphere, whose image winds around the target space once or multiple times. It is indeed known that ferromagnetic skyrmions have nonzero $w[\boldsymbol{n}]$ and satisfy the correct boundary condition, approaching a constant with a $1/|\boldsymbol{x}|^2$ convergence at spatial infinity [27].

Before moving to the next example, it is instructive to contrast the case of ferromagnets to that of antiferromagnets. In the latter, the pattern of breaking global symmetry, $\mathrm{SU}(2)\to\mathrm{U}(1)$, is the same as in the former, but the set of maps $\boldsymbol{n}(\boldsymbol{x}): \mathbb{R}^2 \to S^2$ is now just the configuration space. The phase space consists of maps from $\mathbb{R}^2$ to the four-dimensional noncompact manifold $\mathcal{M}\simeq T^*S^2$, the cotangent bundle of the sphere. Resorting for the moment to local coordinates (NG fields) $\pi^a$ on $S^2$, the symplectic form on $\mathcal{M}$ can be defined likewise locally as $\Omega = \mathrm{d}\Pi_a\wedge\mathrm{d}\pi^a$, where $\Pi_a$ are the conjugate momenta to $\pi^a$. Consider now a set of fields $(\pi^a, \Pi_a)$ satisfying a trivial boundary condition at infinity, thus mapping $\mathbb{R}^2$ to a 2-cycle in $\mathcal{M}\simeq T^*S^2$. We can deform the map smoothly by scaling the conjugate momenta uniformly down to zero. Its image is therefore homologically equivalent to a 2-cycle in the base space of $\mathcal{M}$, that is the sphere $S^2$ itself. However, when projected to the base space, our symplectic form on $\mathcal{M}$ vanishes. We conclude that in antiferromagnets, our would-be topological charge $Q$ vanishes for any field configuration that satisfies a trivial boundary condition at spatial infinity. The same is obviously true for any theory defined by second-order dynamics on a manifold $\mathcal{N}$, whose phase space consists of maps $\mathbb{R}^2\to T^*\mathcal{N}$. The conclusion that $Q$ vanishes also follows directly from the fact that the symplectic form on any cotangent bundle $T^*\mathcal{N}$ is necessarily exact.

The above of course does not imply that there are no topological solitons in antiferromagnets. One can construct antiferromagnetic skyrmion configurations that carry nonzero Brouwer degree as maps $\mathbb{R}^2\to S^2$ just like in ferromagnets. The physics of ferro- and antiferromagnetic skyrmions is however very different. The momentum of the latter does not correspond to the dipole moment of the skyrmion charge. By the same token, their mobility is not restricted by a dipole-type conservation law. Last but not least, the algebra of spatial momentum in antiferromagnets is not centrally extended in the presence of a skyrmion.

## 4.2 Quantum crystals

As the next example, consider a quantum Wigner crystal, expected to be realized for instance in a two-dimensional electron gas subjected to a strong magnetic field $B$. The low-energy EFT

of a Wigner crystal is given by the action [15]

$$S = \frac{n_0}{2\ell^2} \int d^2\boldsymbol{x}\, dt\, \epsilon_{ab} X^a \partial_0 X^b + \cdots. \tag{31}$$

Here $n_0$ is the average particle number density in the ground state, and $\ell \equiv 1/\sqrt{eB}$ the magnetic length. Finally, $X^a(\boldsymbol{x}, t)$ are the Lagrangian coordinates describing the local displacement of the crystal from equilibrium. Shifting the crystal by any lattice vector brings it to the same quantum state, hence the target space is the torus, $\mathcal{M} \simeq T^2$. In order to avoid divergences arising from spatial integration, we restrict to field configurations defined on a single unit cell of the crystal with an implicit periodic boundary condition. This is equivalent to compactifying the domain of the fields $X^a$ to a torus. The phase space then consists of maps $X^a : T^2 \rightarrow T^2$ and the symplectic 2-form can be extracted from the action (31),

$$\Omega = \frac{n_0}{2\ell^2} \epsilon_{ab} dX^a \wedge dX^b. \tag{32}$$

In accord with (18), the topological charge $Q$ is given by

$$Q = \frac{n_0}{2\ell^2} \int_{T^2} d^2\boldsymbol{x}\, \epsilon^{ij} \epsilon_{ab} \partial_i X^a \partial_j X^b = \frac{n_0 S}{\ell^2} = n_0 SeB, \tag{33}$$

where $S$ is the area of the unit cell of the crystal; we used the fact that up to the factor of $n_0/\ell^2$, $\Omega$ is just the area form on $T^2$. It follows that the components of momentum in a Wigner crystal satisfy the Poisson bracket

$$\{P_i, P_j\} = -\epsilon_{ij} n_0 SeB. \tag{34}$$

This agrees with the standard centrally extended algebra of magnetic translations.

Note that in this case, the "topological charge" $Q$ does not actually arise from a specially designed soliton configuration. Rather, the Lagrangian map $X^a(\boldsymbol{x}, t)$ is required to be a smooth invertible deformation of the ground state, $\langle X^a(\boldsymbol{x}, t)\rangle = x^a$, and thus necessarily belongs to the same homotopy class as the latter, which winds around the target space once. The same reasoning as above applies to any quantum crystal whose action includes a Berry term such as (31). A concrete example might be for instance the Abrikosov vortex lattice in rotating superfluids; see [28] for a recent discussion of this system from the fracton point of view.

## 4.3 Superfluids

It has been known for a long time that the components of momentum of a superfluid do not commute in the presence of a vortex [29, 30]. Let us see how to understand this within our framework. First of all, unlike the two-dimensional magnetic skyrmion, the superfluid vortex cannot be realized by a smooth configuration of fields taking values from the coset space of broken symmetry, in this case U(1). This underlines the difference between topological defects (such as vortices) and textures (such as skyrmions). To have a hope for a smooth description of a vortex, we have to take a step back and use a Gross–Pitaevskii-like theory for the complex superfluid condensate $\psi(\boldsymbol{x}, t)$. Its action reads, schematically,

$$S = \int d^2\boldsymbol{x}\, dt\, i\psi^\dagger \partial_0 \psi + \cdots. \tag{35}$$

In this case, the target space is $\mathcal{M} \simeq \mathbb{C}$. The symplectic potential is now $\omega = i\psi^\dagger d\psi$ and the symplectic 2-form reads

$$\Omega = i\, d\psi^\dagger \wedge d\psi. \tag{36}$$

Using (18), we then define the topological charge $Q$ as

$$Q = i \int d^2\boldsymbol{x}\, \epsilon^{ij} \partial_i \psi^\dagger \partial_j \psi \,. \tag{37}$$

The symplectic 2-form (36) is obviously exact, so how can $Q$ ever be nonzero? To that end, recall that a vortex is a topologically nontrivial state such that at spatial infinity, $|\psi(\boldsymbol{x},t)|^2 \equiv n_0$ is fixed and lies on the vacuum manifold, $U(1) \simeq S^1$. Near spatial infinity, we can then parameterize the condensate function by its phase $\theta(\boldsymbol{x},t)$ as $\psi(\boldsymbol{x},t) \to \sqrt{n_0}e^{i\theta(\boldsymbol{x},t)}$. Consequently,

$$Q = i \oint_{\partial \mathbb{R}^2} d\boldsymbol{x} \cdot (\psi^\dagger \boldsymbol{\nabla} \psi) = -n_0 \oint_{\partial \mathbb{R}^2} d\boldsymbol{x} \cdot \boldsymbol{\nabla}\theta \,, \tag{38}$$

where $\partial\mathbb{R}^2$ denotes a large oriented circle at spatial infinity. In other words, the vortex maps the coordinate space $\mathbb{R}^2$ into a disk $D$ in $\mathbb{C}$, whereby the spatial boundary $\partial\mathbb{R}^2$ is mapped to the boundary $\partial D$. While the symplectic 2-form $\Omega$ itself is exact, its potential $\omega$ projects down to a nontrivial generator of the first de Rham cohomology group of $\partial D \simeq S^1$. It is therefore the nontrivial boundary condition, satisfied by the vortex, that makes nonzero $Q$ possible.

From (38) we conclude that $Q = -2\pi\nu n_0$, where $\nu \in \mathbb{Z}$ is the winding number of the superfluid phase. Accordingly, the components of momentum then satisfy

$$\{P_i, P_j\} = 2\pi\epsilon_{ij}\nu n_0 \,. \tag{39}$$

This explains the origin of the dipole symmetry, underlying the fracton-like behavior of superfluid vortices [6].

## 5 Generalization to higher dimensions

In $d > 2$ spatial dimensions, the closed 2-form $\Omega$ on the target space gives rise to a $(d-2)$-form symmetry. The corresponding integral charge is defined by integrating (the pullback of) $\Omega$ over a two-dimensional surface $\Sigma$ in $\mathbb{R}^d$. Our analysis of the algebraic structure of the moments of the topological charge therefore requires modification.

In order to keep the discussion elementary, we restrict the choice of $\Sigma$ to two-dimensional planes spanned by two chosen Cartesian coordinates, $x^I$ and $x^J$. Accordingly, within this section, spatial indices $i,j$ will run over the set $\{I,J\}$. We will now consider a natural subset of Euclidean translations and the Poisson brackets of the corresponding generators. We divide all the Cartesian coordinates in $\mathbb{R}^d$ as $\boldsymbol{x} = (\boldsymbol{y},\boldsymbol{z})$, where $\boldsymbol{z} = (x^I, x^J)$ and $\boldsymbol{y}$ collects all the remaining coordinates. The class of functionals (18) then extends to arbitrary $d$ as

$$Q_\lambda(\boldsymbol{y}) \equiv \int d^2\boldsymbol{z}\, \lambda(\boldsymbol{x}) \epsilon^{ij} \partial_i \omega_j(\phi(\boldsymbol{x})) = \frac{1}{2} \int d^2\boldsymbol{z}\, \lambda(\boldsymbol{x}) \epsilon^{ij} \Omega_{ab}(\phi(\boldsymbol{x})) \partial_i \phi^a(\boldsymbol{x}) \partial_j \phi^b(\boldsymbol{x}) \,. \tag{40}$$

These are functions on $\mathbb{R}^{d-2}$, generating transformations of local fields $\phi^a(\boldsymbol{x})$ in the plane defined by fixed $\boldsymbol{y}$,

$$\delta_{\lambda,\boldsymbol{y}'} \phi^a(\boldsymbol{x}) \equiv \{\phi^a(\boldsymbol{x}), Q_\lambda(\boldsymbol{y}')\} = -\epsilon^{ij} \partial_i \lambda(\boldsymbol{x}) \partial_j \phi^a(\boldsymbol{x}) \delta(\boldsymbol{y}-\boldsymbol{y}') \,. \tag{41}$$

Similarly, the Poisson bracket of two different charges, $Q_\lambda(\boldsymbol{y})$ and $Q_{\bar{\lambda}}(\boldsymbol{y}')$, generalizes straightforwardly (19),

$$\{Q_\lambda(\boldsymbol{y}), Q_{\bar{\lambda}}(\boldsymbol{y}')\} = -Q_{\{\lambda,\bar{\lambda}\}}(\boldsymbol{y}) \delta(\boldsymbol{y}-\boldsymbol{y}') \,. \tag{42}$$

Following the analogy, we next define the generators of in-plane translations and rotations,

$$P_i(\boldsymbol{y}) \equiv Q_{-\epsilon_{ij}z^j}(\boldsymbol{y}), \qquad L(\boldsymbol{y}) \equiv Q_{(1/2)\boldsymbol{z}^2}(\boldsymbol{y}) \,. \tag{43}$$

These satisfy the local Poisson brackets, generalizing (23),

$$\{L(\boldsymbol{y}), P_i(\boldsymbol{y}')\} = \epsilon_i{}^j P_j(\boldsymbol{y})\delta(\boldsymbol{y}-\boldsymbol{y}'), \qquad \{P_i(\boldsymbol{y}), P_j(\boldsymbol{y}')\} = -\epsilon_{ij}Q_1(\boldsymbol{y})\delta(\boldsymbol{y}-\boldsymbol{y}'). \qquad (44)$$

Of course, in-plane translations and rotations are generally not symmetries of the dynamics, so that $P_i(\boldsymbol{y})$ and $L(\boldsymbol{y})$ are generally not conserved. However, if we merely want to focus on the consequences of the central extension of the algebra of infinitesimal translations, then a good starting point is to integrate the second relation in (44) over $\mathbb{R}^{d-2}$ to get

$$\boxed{\{P_i, P_j(\boldsymbol{y})\} = -\epsilon_{ij}Q_1(\boldsymbol{y}).} \qquad (45)$$

Here $P_i$ is the generator of global translations, that is momentum, and (45) is the physically appropriate generalization of (5). Its right-hand side is determined by the integral charge $Q_1(\boldsymbol{y})$ of the $(d-2)$-form symmetry generated by $\Omega$, measured in the $x^I x^J$ plane.

Let us quickly summarize what we have done until now. We have shown the general existence of a local dipole-type conservation law in bosonic field theory, as summarized by (15) and (16). The corresponding charge density is given directly by the symplectic 2-form of the theory, whereas the tensor current descends from the stress tensor. The existence of this local conservation law is reflected by the central extension of the algebra of spatial translations, see (5) and (45). Put all together, these constitute the first main result of our paper. We stress that specific examples of the structure revealed here, including all those discussed in Sec. 4, were known previously in the literature. What is new here is the demonstration of the universality of the dipole conservation law (1) and the central extension of the momentum algebra (5), including their higher-dimensional generalizations (15) and (45), which only relies on translation invariance and the basic symplectic structure of the given theory.

## 6 Linear momentum problem

Suppose that the theory of interest possesses a well-defined momentum density, $p_i(\boldsymbol{x}, t)$. After all, this is expected from translation invariance via Noether's theorem. By the defining property of momentum $P_i$ as the generator of translations, the momentum density must satisfy

$$\delta_i p_j(\boldsymbol{x}) \equiv \{p_j(\boldsymbol{x}), P_i\} = -\partial_i p_j(\boldsymbol{x}). \qquad (46)$$

For a localized, smooth field configuration that decreases sufficiently rapidly at infinity, integration over space then necessarily leads to $\{P_i, P_j\} = 0$. In $d \geq 3$ spatial dimensions, one can integrate just over a two-dimensional subspace and still conclude that $\{P_i, P_j(\boldsymbol{y})\} = 0$ for $i, j$ that label coordinates on the integration subspace. The conditions underlying the vanishing of the integral of the right-hand side of (46) offer two possibilities how $\{P_i, P_j\}$ could actually be nonzero: for fields that either have a singularity or satisfy a nontrivial boundary condition at spatial infinity. One can to some extent switch between these two options by choosing different representations of momentum density [10].

The above said, there are theories where a nonzero value of the charge $Q$ in (5) or $Q_1(\boldsymbol{y})$ in (45) can be realized by fields that are both smooth and sufficiently fast-convergent. The ferromagnetic skyrmion is but one example. In such cases, only one logical possibility remains: a well-defined momentum density $p_i(\boldsymbol{x}, t)$ does not exist. This is the linear momentum problem. It has been known to exist in ferromagnets [31, 32] and superfluid $^3$He for several decades.

Before further developing this idea, let us first see why some obvious candidates for the momentum density cannot do the job. First, naively applying Noether's theorem would suggest that $p_i = -\omega_a \partial_i \phi^a = -\omega_i$. But this may not be globally well-defined on the target space $\mathcal{M}$,

since otherwise the symplectic 2-form $\Omega$ would necessarily be exact, leading to vanishing $Q$. Concretely, this "canonical" momentum density may be ill-defined for field configurations that map $\mathbb{R}^2$, or the two-dimensional plane in $\mathbb{R}^d$ on which $P_i(\boldsymbol{y})$ acts, to a homologically nontrivial 2-cycle in $\mathcal{M}$. Second, guided by (18) and (22), we might instead think of $\tilde{p}_i = -\epsilon_{ij} x^j \epsilon^{kl} \partial_k \omega_l$ as a good candidate. The two would-be momentum densities differ by a mere surface term, as is easily seen from the fact that the tensor current $K^{\mu\nu} \equiv x^\nu J^\mu + \epsilon^{\mu\nu\lambda} \omega_\lambda = \partial_\kappa (x^\nu \epsilon^{\mu\kappa\lambda} \omega_\lambda)$ is conserved off-shell. However, $\tilde{p}_i$ violates the property $\{\phi^a(\boldsymbol{x}), \tilde{p}_i(\boldsymbol{y})\} = -\partial_i \phi^a(\boldsymbol{x}) \delta(\boldsymbol{x} - \boldsymbol{y})$ one would expect from the generator of local translations, due to its explicit dependence on spatial coordinates. Likewise, it violates (46); a short calculation shows that

$$\{\tilde{p}_j(\boldsymbol{x}), P_i\} = -\partial_i \tilde{p}_j(\boldsymbol{x}) + \epsilon_{ij} J^0(\boldsymbol{x}). \tag{47}$$

This is another way to understand the origin of the central extension (5). In a certain sense, the central extension of the momentum algebra by the topological charge $Q$ acts as a "classical anomaly" [14] that obstructs a consistent definition of momentum density.

Now why exactly should that be a problem? Suppose that our action (6) defines a low-energy EFT of some underlying, local continuous microscopic theory. Such a microscopic theory can presumably be coupled to a spacetime background in a way that manifests diffeomorphism invariance. This is certainly true for the atomic-scale description of any condensed-matter system such as those discussed in Sec. 4. A momentum density can then be generated by taking the variation of the action with respect to the appropriate component of the spacetime vielbein. The presence of LMP in the EFT indicates that the EFT cannot be coupled consistently to the same spacetime background as the microscopic theory. The logical conclusion is that the EFT is incomplete: within the energy scale accessible to it, the system must possess further degrees of freedom whose addition to the EFT cures the LMP.

This is our second main result: some bosonic theories cannot be realized as the low-energy EFT of any microscopic, translationally invariant local field theory. The presence of LMP depends only on the topology of the target space $\mathcal{M}$ and the choice of the symplectic 2-form $\Omega$ on it, regardless of the number $d$ of spatial dimensions.

## 7 Implications for the spectrum of Nambu–Goldstone bosons

The prediction of additional degrees of freedom is particularly striking for EFTs of systems with a spontaneously broken symmetry, whose degrees of freedom are the corresponding NG bosons. Here we can in principle make the cutoff of the EFT arbitrarily small. It then follows that the missing degrees of freedom that must be present to cure the LMP must be gapless as well. However, the mechanism guaranteeing the existence of additional light degrees of freedom is robust due to its topological nature. It will therefore survive even in presence of small perturbations breaking the symmetry that give the NG bosons a gap. This is important since, for instance, in real ferromagnetic materials, the SU(2) symmetry under spin rotations is explicitly broken by crystal anisotropy and spin-orbit coupling.

### 7.1 Forbidden patterns of symmetry breaking

We will now specify more concretely a class of EFTs that feature LMP and thus are, by the above argument, incomplete. We will focus on EFTs for NG bosons of spontaneously broken internal symmetry, whose structure is well-understood [33–35]. Spontaneous breakdown of an internal symmetry group $G$ to a subgroup $H$ gives rise to a set of NG fields, $\pi^a(\boldsymbol{x}, t)$, parameterizing the coset space $G/H$. Assuming spacetime translation invariance and spatial rotation invariance,

the relevant part of the effective Lagrangian for the NG fields is

$$\mathcal{L} = c_a(\pi)\partial_0\pi^a + \cdots,\tag{48}$$

where the ellipsis stands for operators with more than one (spatial or temporal) derivative. The locally defined 1-form $c(\pi) \equiv c_a(\pi)\mathrm{d}\pi^a$ on $G/H$ is constrained by the conditions that $\mathrm{d}c(\pi)$ is closed and $G$-invariant. This allows for the possibility that the Lagrangian itself is invariant only up to a total derivative. It was shown in [10] that whenever $G/H$ is compact and $\mathrm{d}c(\pi)$ is cohomologically nontrivial, the central extension in (5) can be realized by smooth fields that map $\mathbb{R}^2$ to a 2-cycle in $G/H$. The presence or absence of the LMP can therefore be checked by studying the second de Rham cohomology of $G/H$. Vanishing of the second cohomology group guarantees the absence of LMP. Otherwise, it is necessary to check explicitly whether or not $\mathrm{d}c(\pi)$ is exact.

To be even more concrete, we need some further assumptions on the symmetry. Let $G$ be compact, semisimple and simply connected, and let $H$ be connected. Denote as $U(\pi) \in G$ the representative of the coset in $G/H$ with coordinates $\pi^a$. Then the 1-form $c(\pi)$ becomes [33]

$$c(\pi) = \sigma_A[\mathrm{i}U(\pi)^{-1}\mathrm{d}U(\pi)]^A,\tag{49}$$

where $A$ labels the generators $T_A$ of $G$ and $\sigma_A$ is the expectation value of the density of $T_A$ in the ground state. The set of constants $\sigma_A$ establishes a vector in the adjoint representation of $G$. By the $H$-invariance of the ground state, this vector carries a trivial one-dimensional representation (singlet) of $H$. It is however only those singlets that lie entirely in the Lie algebra of $H$ that can make $\mathrm{d}c(\pi)$ cohomologically nontrivial. Mathematically, the generators of the second de Rham cohomology of $G/H$ are in a one-to-one correspondence with the U(1) factors of $H$ [36].

In plain terms, under our assumptions that $G$ is compact, semisimple and simply connected and $H$ is connected, the EFT for NG bosons exhibits LMP if and only if some *unbroken* generators of $G$ have a nonzero density in the ground state. A prominent example of such a system is, of course, the ferromagnet. Here, $G/H \simeq \mathrm{SU}(2)/\mathrm{U}(1)$ and the ground state carries nonzero density of the unbroken U(1) generator. The spin density itself acts as the order parameter for symmetry breaking.

The situation where the ground state is an eigenstate of an order parameter commuting with the Hamiltonian is very special (see [37] for some further details) and has an interesting history. Three decades ago, Anderson [38] argued that ferromagnets are not an example of spontaneous symmetry breaking, not of the type seen in particle physics at least. He proposed that in the latter, there are no theories with spontaneously broken symmetry of the "conserved type." We can now see why: no consistent Lorentz-invariant field theory can have a low-energy EFT description in terms of the ferromagnetic SU(2)/U(1) coset space, without any additional gapless degrees of freedom. This observation extends to the whole class of symmetry-breaking patterns satisfying our assumptions on $G$ and $H$ where some generator of $H$ has a nonzero vacuum expectation value.

## 7.2 Completing the low-energy effective theory

Suppose we are given a low-energy EFT that exhibits the LMP, but we do not know the details of the underlying microscopic theory. Then there is one logical possibility that we have not mentioned until now: that the microscopic theory simply does not possess a local momentum conservation law. In such cases, the translation invariance of the EFT is an emergent property, only valid at sufficiently long distances. Should we on the other hand know that the microscopic theory has continuous translation invariance, we can deduce the existence of additional gapless degrees of freedom. These can be fermionic; a natural possibility is the presence of a

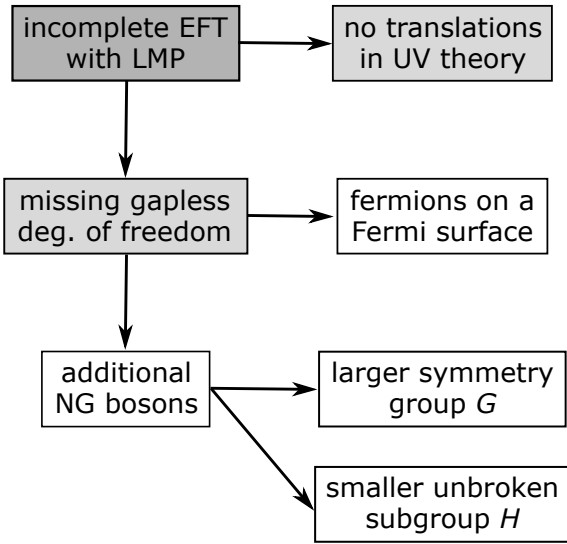

Figure 2: Visualization of the different possibilities implied by the presence of the LMP in a low-energy EFT for NG bosons. Unless the microscopic (UV) theory does not have continuous translation invariance, further gapless degrees of freedom must be present. Generically, these can be either gapless fermions or additional NG bosons, stemming from an enlarged coset space.

gapless Fermi sea. For bosons, the only robust mechanism guaranteeing the absence of a gap is via spontaneous symmetry breaking. The additional NG bosons then belong to an enlarged coset space $G'/H'$, where either $G' \supsetneq G$ or $H' \subsetneq H$, or both. For the reader's convenience, we display the various possibilities graphically in Fig. 2.

Intriguingly, all the different possibilities can be realized in ferromagnets. First, there are lattice models of ferromagnetism that are exactly solvable so that one can explicitly demonstrate that ferromagnetic spin waves (magnons) are the only gapless modes in the spectrum. It was shown already in mid-1980s by Haldane [31] that in the continuous low-energy EFT of such models, the integral momentum has a topological ambiguity. This ambiguity leaves only operators of certain discrete translations well-defined, namely those corresponding exactly to the translation symmetry of the underlying lattice. In this case, the absence of a well-defined momentum density is not a problem, but rather should be expected.

Real ferromagnets, however, should have a continuous translationally invariant description at the atomic scale. We are thus led to the prediction of additional gapless modes in any ferromagnetic material. Now most natural ferromagnets are metallic, hence include gapless itinerant electrons. Volovik [32] showed that while the momentum density is ill-defined separately for the magnon and electron subsystems, the LMP disappears when the two subsystems are put together. Moreover, there is another, related reason why the EFT for spin waves alone is unsatisfactory. Galilei invariance requires that momentum transfer in uniform metallic ferromagnets is accompanied by en electric current. This requires in turn a nonlocal coupling of magnons to the electromagnetic field [39]. The problem is again cured by adding the electron degrees of freedom. This is an example of a completion of the EFT by gapless fermions. Another example of this type is the A-phase of superfluid ${}^3$He. Here the NG sector features a LMP, which is cured by the presence of gapless nodes on the Fermi surface [40].

Ferromagnetic insulators are arguably not as common as ferromagnetic metals, but do exist in nature as well. Here the resolution of the LMP must arise from additional gapless bosons. It appears that the only possible candidates are the phonons arising from spontaneous breaking

of translations by the underlying crystal lattice. This is an example of curing the LMP by enlarging the symmetry group $G$. It is not immediately obvious why phonons should cure a problem that arises from the topology of the spin coset space. We therefore demonstrate this explicitly in Appendix A. As far as we know, the prediction of a particular magnon–phonon coupling, required to eliminate the LMP in the magnon sector, is new. Our proposal is close in spirit to [41], which argued that the phase structure of ferromagnets is very similar to that of incompressible fluids, and that the LMP can be compensated by adding hydrodynamic variables that allow for density variations.

Finally, there is one simple possibility, corresponding to breaking the residual $H \simeq U(1)$ symmetry. The resulting coset space is $G'/H' \simeq SU(2)/\{e\}$; there are no unbroken generators at all, and hence no LMP. This symmetry-breaking pattern is realized in canted ferromagnets.

This last example is a representative of a class of EFT completions that occur naturally in systems with nonrelativistic NG bosons, whenever the internal symmetry group $G$ is compact and semisimple. Suppose that out of all the order parameters needed to break $G$ down to $H$, we only keep the charge densities in the ground state. It is possible to choose a basis of the Lie algebra of $G$ so that only mutually commuting generators have a nonzero expectation value [42]. The charge density order parameters then generate certain Abelian subgroup (torus) $T \subset G$. At the same time, they break the symmetry under $G$ down to $K = \{g \in G \,|\, gh = hg \;\forall h \in T\}$, that is the centralizer of $T$ in $G$. The coset space $G/K$ accommodates all the *type-B* NG bosons [43] of the EFT. This provides an extreme example of LMP, since all charge densities in the ground state are by construction unbroken. The 2-form $\Omega_{G/K} = dc_{G/K}(\pi)$ on $G/K$ defines a symplectic structure, which ensures that the number of independent type-B NG modes is $(1/2) \dim G/K$.

Now we add whatever other order parameters are needed to break $K$ further down to $H$. This results in additional $\dim K/H$ *type-A* NG bosons in the spectrum. The coset spaces $G/K$ and $K/H$ are geometrically collated, giving rise to a fiber bundle structure on $G/H$ [33],

$$K/H \to G/H \xrightarrow{\pi} G/K \,. \tag{50}$$

The projection $\pi : G/H \to G/K$ amounts to neglecting the type-A modes. In canted ferromagnets, one has respectively $K/H \simeq U(1)/\{e\}$, $G/H \simeq SU(2)/\{e\}$ and $G/K \simeq SU(2)/U(1)$. Embedding the base space $G/K$ in the larger manifold $G/H$ will typically reduce the second de Rham cohomology group. In physics terms, this means that the LMP present in the type-B NG sector is partially or completely compensated by adding the type-A NG degrees of freedom.

The above discussion of LMP in EFTs for NG bosons was based on the geometry of the coset space $G/H$. This is in contrast to the rest of our paper, where the target space $\mathcal{M}$ of the phase-space field variables plays the key role. The apparent discrepancy can be addressed by extending the fiber bundle structure (50) to

$$T^*(K/H) \to \mathcal{M} \xrightarrow{\tilde{\pi}} G/K \,, \tag{51}$$

where the projection $\tilde{\pi}$ now amounts to ignoring the type-A NG fields as well as their conjugate momenta. Applying the same reasoning we used in Sec. 4.1 to deal with antiferromagnets, we can now deform any 2-cycle in $\mathcal{M}$ to a 2-cycle in $G/H$. When projected to the latter, the symplectic form on $\mathcal{M}$ reduces to $\pi^* \Omega_{G/K} = dc_{G/H}$. This explains why the possible presence of the LMP in EFTs of NG bosons can be inspected by looking at the second de Rham cohomology group of $G/H$ and the properties of the 2-form $dc(\pi)$, defined by (48).

# 8 Nambu–Goldstone bosons of higher-form symmetries

The main conclusion of Sec. 7 that the LMP requires the presence of additional gapless degrees of freedom, extends to EFTs for NG bosons of spontaneously broken higher-form sym-

metry [44–46]. While we leave a more detailed and general analysis to future work, we wish to give here at least one concrete, physically relevant example.

Consider Maxwell's electrodynamics in $d = 3$ spatial dimensions, coupled to a background axion-like field $\theta$. Its Lagrangian is given in terms of the electric and magnetic fields $\boldsymbol{E}, \boldsymbol{B}$ as

$$\mathscr{L} = \frac{1}{2}(\boldsymbol{E}^2 - \boldsymbol{B}^2) + C\theta \boldsymbol{E} \cdot \boldsymbol{B}. \tag{52}$$

For the time being, we treat $\theta$ as an arbitrary but fixed background that is static but may depend on the spatial coordinates $\boldsymbol{x}$. The coupling $C$ can then be treated as a continuously tunable parameter. For $C = 0$, the theory (52) is just free electrodynamics. It describes the photon as the NG boson of a spontaneously broken 1-form symmetry [44]. The case of $C \neq 0$ (which we henceforth assume without explicitly saying so) is more interesting, as it features a dramatically modified low-energy spectrum. To demonstrate this, we switch to the temporal gauge in which the scalar potential is set to zero. This leaves us with a residual invariance under time-independent gauge transformations of the vector potential $\boldsymbol{A}$. Upon integration by parts at the level of the action, the Lagrangian (52) can be rewritten as

$$\mathscr{L} = \frac{1}{2}(\partial_0 \boldsymbol{A})^2 - \frac{1}{2}(\boldsymbol{\nabla} \times \boldsymbol{A})^2 + \frac{C}{2}\boldsymbol{\nabla}\theta \cdot (\boldsymbol{A} \times \partial_0 \boldsymbol{A}). \tag{53}$$

Of particular interest is the special case where $\boldsymbol{\nabla}\theta$ is a periodic function of coordinates with nonzero spatial average. The spectrum of photon excitations then has a band structure similarly to crystalline solids. Due to the presence of the term with a single time derivative, one of the photon helicity eigenstates acquires a gap. The other helicity eigenstate is softened, its energy being proportional to squared momentum [47–49]. This is an example of a type-B NG boson of higher-form symmetry [50].

The axion electrodynamics (52) is known to have an intricate symmetry structure [51,52]. However, here we focus solely on the interplay of gauge invariance and spatial translations. This is nontrivial, as follows by inspection of (52) and (53). While (52) is manifestly gauge-invariant, it breaks translations for any non-constant $\theta$. Likewise, (53) may preserve translations if $\theta$ has a constant gradient at the cost of sacrificing manifest gauge invariance. The same dichotomy appears if we try to define a momentum density. The naive momentum density given by the Poynting vector, $\boldsymbol{p}_0 = \boldsymbol{E} \times \boldsymbol{B}$, is gauge-invariant but no longer satisfies a local conservation law. The axion background $\theta(\boldsymbol{x})$ exerts a force on the electromagnetic field with density $C\boldsymbol{\nabla}\theta(\boldsymbol{E} \cdot \boldsymbol{B})$. In the special case that $\boldsymbol{\nabla}\theta$ is constant, one can absorb this force into a redefinition of momentum density,

$$\boldsymbol{p}_\theta = \boldsymbol{E} \times \boldsymbol{B} + \frac{C}{2}\boldsymbol{\nabla}\theta(\boldsymbol{A} \cdot \boldsymbol{B}). \tag{54}$$

Its spatial integral, $\boldsymbol{P}_\theta = \int \mathrm{d}^3\boldsymbol{x}\, \boldsymbol{p}_\theta(\boldsymbol{x}, t)$, is then conserved. The price to pay is that $\boldsymbol{p}_\theta$ is no longer gauge-invariant although $\boldsymbol{P}_\theta$ itself is, at least under transformations $\boldsymbol{A} \to \boldsymbol{A} + \boldsymbol{\nabla}\lambda$ such that the gauge parameter $\lambda(\boldsymbol{x})$ vanishes sufficiently rapidly at spatial infinity. This is not a consequence of mere bad choice. In presence of the axion background, there is no gauge-invariant momentum density whose spatial integral could serve as a generator of spatial translations. We provide a detailed proof of this claim in Appendix B.

This is the essence of the LMP in axion electrodynamics. Its resolution can follow all the three qualitatively different paths suggested in Sec. 7.2 and summarized in Fig. 2. First, we may insist that the background $\theta(\boldsymbol{x})$ is given externally, in which case the theory (52) simply is not translationally invariant in the direction of $\boldsymbol{\nabla}\theta$. The absence of a well-defined momentum density is then hardly surprising.

Alternatively, we may want to realize $\theta(\boldsymbol{x})$ dynamically as a condensate of some dynamical pseudoscalar degree of freedom. One explicit realization of such a condensate appears in

quantum chromodynamics (QCD) as the so-called chiral soliton lattice state [53,54]. Here $\theta$ represents the neutral pion, which in presence of a baryon chemical potential and a uniform magnetic field $\boldsymbol{B}$ develops a condensate that is quasi-periodic in the direction of $\boldsymbol{B}$. Uniform $\boldsymbol{\nabla}\theta$ requires taking the chiral limit of vanishing quark, and hence pion, masses. By the same token, the spectrum of pseudoscalar excitations above such a background is then gapless. The neutral pion itself therefore supplies the additional NG mode required by the LMP.

Another realization of the axion-like background with nonzero $\boldsymbol{\nabla}\theta$ is through the so-called meson supercurrent phase [55,56], which may appear in dense QCD due to Fermi surface splitting induced by the relatively large strange quark mass. The spatially-varying meson condensate is accompanied by gapless fermions [57,58]. In fact, the fermions carry a current that completely compensates the meson supercurrent. This is required by the Bloch theorem for relativistic systems [59], which forbids any current in the ground state in the thermodynamic limit. This is yet another mechanism how the LMP can be resolved by the presence of additional gapless degrees of freedom.

Before we close the discussion of axion electrodynamics, we remark that it also possesses a natural dipole-type conservation law where both the charge density and the tensor current are gauge-invariant. Start by noting that the closed 3-form $d\theta \wedge dA$ is the Hodge dual of a topological current whose temporal and spatial components are, up to arbitrary normalization,

$$J^0 = -C\boldsymbol{B}\cdot\boldsymbol{\nabla}\theta, \qquad \boldsymbol{J} = -C\boldsymbol{E}\times\boldsymbol{\nabla}\theta. \tag{55}$$

These are respectively the electric charge density and current, carried by the axion background, as follows from the modified Maxwell equations

$$\boldsymbol{\nabla}\cdot\boldsymbol{E} = -C\boldsymbol{B}\cdot\boldsymbol{\nabla}\theta, \qquad \boldsymbol{\nabla}\times\boldsymbol{B} = \partial_0\boldsymbol{E} - C\boldsymbol{E}\times\boldsymbol{\nabla}\theta. \tag{56}$$

While these equations of motion are valid for any function $\theta(\boldsymbol{x})$ of spatial coordinates, suppose now that $\boldsymbol{\nabla}\theta$ is constant, and represent its direction by a unit vector, $\boldsymbol{n}$. By projecting the modified Ampère law in (56) to $\boldsymbol{n}$ and taking another derivative, we get

$$\partial_0(\boldsymbol{\nabla}\cdot\boldsymbol{E}_\parallel) - (\boldsymbol{n}\cdot\boldsymbol{\nabla})[\boldsymbol{n}\cdot(\boldsymbol{\nabla}\times\boldsymbol{B})] = 0, \tag{57}$$

where $\boldsymbol{E}_\parallel \equiv (\boldsymbol{n}\cdot\boldsymbol{E})\boldsymbol{n}$ is the component of $\boldsymbol{E}$ parallel to $\boldsymbol{\nabla}\theta$. In the low-energy limit where only the nonrelativistic type-B photon mode is active, the Ampère law dictates that the part of $\boldsymbol{E}$ perpendicular to $\boldsymbol{\nabla}\theta$ scales as a derivative of $\boldsymbol{B}$ and is small. We then expect the divergence $\boldsymbol{\nabla}\cdot\boldsymbol{E}$ to be dominated by the longitudinal component $\boldsymbol{E}_\parallel$. By the modified Gauss law in (56), the first term in (57) is then well approximated by $-\partial_0(C\boldsymbol{B}\cdot\boldsymbol{\nabla}\theta)$. We thus arrive at a dipole-type conservation law for the magnetic field, valid in the low-energy limit,

$$\partial_0(C\boldsymbol{B}\cdot\boldsymbol{\nabla}\theta) + (\boldsymbol{n}\cdot\boldsymbol{\nabla})[\boldsymbol{n}\cdot(\boldsymbol{\nabla}\times\boldsymbol{B})] \approx 0. \tag{58}$$

The corresponding scalar charge density matches the topological charge in (55).

## 9 Discussion and Conclusions

In this paper we have pointed out the general existence of a local dipole conservation law, the only assumption being continuous translation invariance and absence of higher time derivatives in the Lagrangian. The two main ingredients entering the dipole conservation law are the density of a topological charge and the stress tensor. The topological charge, descending from the symplectic form of the system, gives rise to a central extension of the algebra of spatial momentum. This seems to be a particular realization of a more general phenomenon pointed

out recently by Nair [60], whereby certain topologically nontrivial (Wess–Zumino) terms in the action may lead to anomalous commutators of the energy–momentum tensor.

When the symplectic form is cohomologically nontrivial, nonzero topological charge may be generated by smooth field configurations that converge to a constant at spatial infinity. This implies the presence of LMP: the theory does not possess a well-defined momentum density. We argued that in case the short-distance physics of the system is captured by a local, translationally invariant microscopic theory, a would-be low-energy EFT featuring the LMP is necessarily incomplete, indicating the presence of additional degrees of freedom. This is reminiscent of the Berry phase [61], which arises in quantum mechanics upon integrating out a set of heavy degrees of freedom. However, the additional modes compensating for the LMP must appear at an energy scale accessible to the EFT. This resembles more the constraints on the low-energy spectrum, imposed by 't Hooft anomalies [62].

The consequences of the LMP are particularly striking when it appears in a low-energy EFT for NG bosons of a spontaneously broken global symmetry. Namely, the missing degrees of freedom must then be gapless just like the NG bosons themselves. This gives rise to a no-go theorem, forbidding certain symmetry-breaking patterns in any microscopic, translationally invariant local field theory. Two generic options for the additional gapless modes required by the LMP are fermions on a gapless Fermi surface, and further NG modes. The intimate relation between the LMP and the presence of additional gapless modes in the spectrum seems to extend beyond the class of systems with scalar NG bosons. As we showed on a concrete example, the same happens in theories of NG bosons of higher-form symmetries.

Let us close the paper with some concluding remarks. First, suppose that the action (6) is merely an approximation to a more complete theory, singled out for instance by the classical limit or as the leading order in a derivative expansion. How are the results of this paper going to change when higher-order corrections are added to the action? Should the correction consist solely of operators without time derivatives, it can always be absorbed into the Hamiltonian $\mathcal{H}$. This implies a redefinition of the stress tensor $\sigma_{ij}$ and the spatial tensor current $J^{i_1\cdots i_{d-2}jk}$ but no further changes. Suppose now that we add to the action a generic operator of higher order in derivatives. We can still write the action in the form (6), except that the "Hamiltonian density" $\mathcal{H}[\phi]$ can now also contain operators with time derivatives. It turns out that the argument proving the existence of a dipole-type conservation law still goes through with just a minor modification. Namely, the argument of the last integral in (7) has to be replaced with $\partial_\mu \xi^j \sigma^\mu{}_j$. Likewise, the right-hand side of (9) becomes $-\partial_\mu \sigma^\mu{}_i$. The part herein containing a time derivative can eventually be absorbed into a redefinition of the charge density $\rho^{i_1\cdots i_{d-2}}$, and one again ends up with a conservation law of the type (15). This makes it clear that the assumption of absence of higher-order time derivatives is in fact not essential for the derivation of the local dipole conservation law. It is only needed in Sec. 3 when we analyze the algebra of conserved charges using the symplectic formalism. All we need for the dipole conservation law itself is that the theory is translationally invariant and that the charge density and current in (15) are well-defined on the target space of the theory. The latter assumption may be violated for instance in theories with a gauge redundancy, as we demonstrate in Sec. 8.

For a concrete illustration of the effects due to perturbations of the Lagrangian, let us consider the presumably dominant correction with time derivatives, which is of the form $\delta S = \int \mathrm{d}^d\boldsymbol{x}\,\mathrm{d}t\,(1/2)g_{ab}(\phi)\partial_0\phi^a\partial_0\phi^b$. In order that the energy of the theory remains bounded from below and mathematically consistent, $g_{ab}(\phi)$ should correspond to a Riemannian metric, globally well-defined on $\mathcal{M}$. Repeating the steps in Sec. 2, one finds that (9) changes to

$$\partial_0\omega_i - \partial_i\omega_0 = -\partial_j\sigma^j{}_i + \frac{1}{2}\partial_i(g_{ab}\partial_0\phi^a\partial_0\phi^b) - \partial_0(g_{ab}\partial_0\phi^a\partial_i\phi^b). \tag{59}$$

The form of the dipole conservation law (1) is then preserved if one replaces $J^0 = \epsilon^{ij}\partial_i\omega_j$ by

$$\tilde{J}^0 = \epsilon^{ij}\partial_i(\omega_j + g_{ab}\partial_0\phi^a\partial_j\phi^b) = -\epsilon^{ij}\partial_i p_j, \tag{60}$$

where $p_i$ is the canonical (Noether) momentum density derived from the perturbed action.

    This points to a simpler way to understand the origin of the dipole conservation law (1). In any translationally invariant field theory, local momentum conservation will take the form $\partial_0 p_i = \partial_j\sigma^j{}_i$, where $p_i$ and $\sigma_{ij}$ are momentum density and stress tensor obtained via Noether's theorem. Upon taking the curl, we get immediately (1), where $J^0 = -\epsilon^{ij}\partial_i p_j$ and $J^{ij}$ is given by (13). A natural question then arises as to whether the class of dipole symmetries discussed in this paper is a trivial consequence of translation invariance without any further content. However, as we point out in Sec. 6, there are theories where a consistent momentum density does not exist, namely those featuring LMP. The dipole conservation law (1) is then more fundamental than local momentum conservation, and its shortcut derivation by taking the curl of the latter should be viewed as a mere mnemonic. What the argument leading to (60) does for us is show that the value of the integral charge $Q$ and the identification (4) of integral momentum $P_i$ with the dipole moment $D^i$ of the topological charge density are robust and unaffected by perturbations of the action that are globally well-defined on $\mathcal{M}$. This also applies to theories without LMP where a nonzero value of $Q$ arises from field configurations satisfying a nontrivial boundary condition, as we saw in Sec. 4.3 on the example of superfluids. Finally, there are theories where a consistent local momentum density exists and the scalar charge $Q$, whenever well-defined and finite, vanishes. In this (arguably "default") case, the local conservation of momentum is primary and the dipole conservation law (15) its descendant, without additional content.

    Our second remark is that in contrast to everything we have written so far, there is in fact a class of perturbations that can break the local dipole-type conservation law explicitly. Namely, the conservation of the topological charge $Q$ can be violated by coupling the system to a gauge field that serves as a source for the defects carrying the charge. (See, e.g., [63] for a related discussion.) For concreteness, we consider the superfluid discussed in Sec. 4.3, where the topological charge $Q$ is the winding number of the superfluid phase. Here the conservation of the topological charge $Q$ can be violated by gauging the U(1) 0-form symmetry dynamically. We introduce a 1-form gauge field $A = A_\mu dx^\mu$, and couple the complex superfluid field $\psi$ to it by replacing ordinary derivatives $\partial_\mu\psi$ with covariant derivatives $D_\mu\psi \equiv \partial_\mu\psi - iA_\mu\psi$. The gauge transformation laws are given by $\psi \to e^{i\Lambda}\psi$ and $A_\mu \to A_\mu + \partial_\mu\Lambda$ with the gauge parameter $\Lambda$. This covariant derivative gives in turn the Stückelberg coupling $\sqrt{n_0}(\partial_\mu\theta - A_\mu)$ of the phase $\theta$ to the gauge field, at low energies where the fluctuations of the absolute value of $\psi$ can be neglected. We remark that in the presence of the U(1) gauge field, the phase $\theta$ should be understood as a superconducting rather than superfluid phase. We will now show that in presence of the gauge field, the dipole symmetry of the superfluid is explicitly broken. There are at least two ways to see this. One is that the topological charge is not gauge invariant. Alternatively, the charge can be modified in a gauge invariant way, but the modified charge is not conserved. For the former case, the topological charge defined by (38) is gauge-transformed as $Q \to Q - n_0\oint_{\partial\mathbb{R}^2} d\boldsymbol{x}\cdot\boldsymbol{\nabla}\Lambda$, and the deviation is nonzero if $\Lambda$ has a winding number. Therefore, the charge $Q$ is not physical. For the latter case, we can modify the charge (38) as $Q^{\mathrm{cov}} \equiv i\int_{\mathbb{R}^2} d^2\boldsymbol{x}\,\epsilon^{ij}(D_i\psi)^\dagger D_j\psi = i\oint_{\partial\mathbb{R}^2} d\boldsymbol{x}\cdot\psi^\dagger(\boldsymbol{\nabla}-i\boldsymbol{A})\psi + \frac{1}{2}\int_{\mathbb{R}^2} d^2\boldsymbol{x}\,\epsilon^{ij}F_{ij}\psi^\dagger\psi$. The first term $i\oint_{\partial\mathbb{R}^2} d\boldsymbol{x}\cdot\psi^\dagger(\boldsymbol{\nabla}-i\boldsymbol{A})\psi$ vanishes by the boundary condition on the vortex solution, $(\boldsymbol{\nabla}-i\boldsymbol{A})\psi = 0$, at $\partial\mathbb{R}^2$. But the second term $\int_{\mathbb{R}^2} d^2\boldsymbol{x}\,\epsilon^{ij}F_{ij}\psi^\dagger\psi$ does not vanish in general. The time derivative of the modified charge may be nonzero, depending on the time evolution of the magnetic field $F_{ij}$ and the electric charge $\psi^\dagger\psi$, which are nonzero in general.

    Third, it is known that in EFTs for NG bosons of a spontaneously broken internal symmetry, nonzero density of an unbroken generator of $G$ results in the 1-form $c(\pi)$ in (48) not being

invariant under $G$ [64]. Specifically, a global transformation from $G$ acts on $c(\pi)$ effectively as an $H$-valued gauge transformation. Accordingly, the Lagrangian density of the EFT as well as its canonical momentum density are not $G$-invariant as well, but merely quasi-invariant (invariant up to a surface term). The fact that momentum density should not be invariant under an *internal* symmetry of the system is certainly unexpected. We therefore stress that the LMP means more than that, indicating absence of any momentum density, $G$-invariant or not.

For further insight, note that quasi-invariant 1-forms $c(\pi)$ are classified by the second Lie algebra cohomology of $G$ relative to $H$. When $G$ is compact and connected and $H$ is closed and connected, this relative Lie algebra cohomology is isomorphic to the second de Rham cohomology of $G/H$ [65]. In this case, the quasi-invariance property of $c(\pi)$ is therefore equivalent to the presence of LMP. For an example where the two properties do not coincide, consider the coset spaces $G/\{e\}$ with $G$ either $U(1) \times U(1)$ or $\mathbb{R} \times \mathbb{R}$. In both cases, the second Lie algebra cohomology is one-dimensional, and the corresponding 1-form $c(\pi) = (1/2)\epsilon_{ab}\pi^a d\pi^b$ is quasi-invariant under $G$, which acts on $\pi^a$ by mere translations. However, the second de Rham cohomology is nontrivial only in the case of $U(1) \times U(1) \simeq T^2$. Here $dc(\pi)$ is not exact, being proportional to the area form on $T^2$. The theory exhibits LMP, since $\pi^a$ are mere local coordinates on the torus. In case of $\mathbb{R} \times \mathbb{R} \simeq \mathbb{R}^2$, on the other hand, $\pi^a$ are globally defined and so is therefore the momentum density. Here the LMP is absent even if the momentum density is not $G$-invariant. This latter case is similar to superfluid vortices, discussed in Sec. 4.3, where the target space within a Gross–Pitaevskii-like approach is $\mathcal{M} \simeq \mathbb{C} \simeq \mathbb{R}^2$.

The generality of some of the results reported in this paper raises a number of questions. Are there other examples of nested conservation laws than those sketched in Fig. 1? Is it possible to recover in a similar way higher-order multipole conservation laws? Another natural avenue for future work would be to explore the general physical consequences of the LMP. We have predicted a very specific coupling between certain NG bosons and additional degrees of freedom, required to make momentum density mathematically consistent. This coupling can be interpreted in terms of a mutual force between the two subsystems, known in metallic ferromagnets as the spin-motive force. The existence of a similar force between the electromagnetic field and the axion background in axion electrodynamics was pointed out in Sec. 8. Finding other examples of the same phenomenon would be extremely interesting.

## Acknowledgments

We are indebted to Grigory Volovik who informed us, years ago, about the linear momentum problem, and to Sergej Moroz for collaboration in an early stage of this project. We would also like to thank Sergej Moroz, Helge Ruddat, Eirik Eik Svanes and Jasper van Wezel for helpful discussions.

**Funding information** The work of T. B. has been supported in part by the grant no. PR-10614 within the ToppForsk-UiS program of the University of Stavanger and the University Fund. The work of N. Y. has been supported in part by the Keio Institute of Pure and Applied Sciences (KiPAS) project at Keio University and JSPS KAKENHI Grant Numbers JP19K03852 and JP22H01216. The work of R. Y. has been supported by JSPS KAKENHI Grant Numbers JP21J00480 and JP21K13928.

# A  Avoiding the linear momentum problem in ferromagnets

Here we illustrate how the LMP is cured by the presence of additional gapless degrees of freedom, using ferromagnetic insulators as an example. Suppose that the ferromagnetic order is carried by a medium whose collective motion contributes additional gapless modes to the spectrum. In order to have a well-defined local low-energy EFT, these collective degrees of freedom should be considered alongside magnons. The local displacement of the medium is described by the Lagrangian coordinates $X^a(\boldsymbol{x}, t)$ where $a = 1, \ldots, d$. Its motion is captured by the following current [66],

$$J_X^\mu \equiv \frac{n(X)}{d!} \epsilon^{\mu \nu_1 \cdots \nu_d} \epsilon_{a_1 \cdots a_d} \partial_{\nu_1} X^{a_1} \cdots \partial_{\nu_d} X^{a_d} . \tag{61}$$

The function $n(X)$ is the density of the medium in the comoving frame. It is fixed by the ground state, and is constant in case the equilibrium state of the medium is uniform. The current (61) is conserved off-shell for any choice of $n(X)$. Its temporal part, $J_X^0 = n(X) \det\{\partial_i X^a\}_{i,a=1}^d$, is the density of the medium in the "laboratory" frame as defined by the coordinates $\boldsymbol{x}$. The spatial part then carries information about the local (Eulerian) velocity $\boldsymbol{v}$ of the medium, $J_X^i = J_X^0 v^i$.

The crystalline order of ferromagnetic insulators gives rise to the additional NG modes, needed to cure the LMP: the phonons. An EFT describing the interacting magnon–phonon system in (anti)ferromagnetic and ferrimagnetic insulators has been developed only recently [67]. Since we assumed the ferromagnetic ground state to be uniform, we set $n(X) = n_0$, where $n_0$ is the particle number density in equilibrium. The relevant part of the magnon action is then affected by the coupling to phonons as follows,

$$\begin{aligned}
S &= \int \mathrm{d}^d \boldsymbol{x} \, \mathrm{d}t \, \omega_a(\phi) \partial_0 \phi^a + \cdots \to \int \mathrm{d}^d \boldsymbol{x} \, \mathrm{d}t \, (\det\{\partial_i X^a\}_{i,a=1}^d) \omega_a(\phi)(\partial_0 + \boldsymbol{v} \cdot \boldsymbol{\nabla}) \phi^a + \cdots \\
&= \frac{1}{n_0} \int \mathrm{d}^d \boldsymbol{x} \, \mathrm{d}t \, \omega_a(\phi) J_X^\mu[X] \partial_\mu \phi^a + \cdots = \frac{1}{d!} \int \epsilon_{a_1 \cdots a_d} \omega(\phi) \wedge \mathrm{d}X^{a_1} \wedge \cdots \wedge \mathrm{d}X^{a_d} + \cdots .
\end{aligned} \tag{62}$$

Why does this eliminate the LMP? A quick calculation shows that the canonical momentum density arising from this term in the action is now well-defined. Namely, it is identically zero. This is most obvious from the last expression in (62), which is a strictly topological term, independent of whatever spacetime background might be present. The momentum of the EFT comes entirely from the phonon sector, and is thus unaffected by the nontrivial topology of the ferromagnetic coset space. This makes perfect sense: it is the phonon rather than magnon fields that are responsible for transport of matter.

In deriving (62), we have not used any properties of the symplectic potential $\omega(\phi)$, specific to ferromagnets. Therefore, the same mechanism may in principle be used to eliminate the LMP from any EFT, only subject to the assumption that the gapless collective modes parameterized by the fields $X^a(\boldsymbol{x}, t)$ are actually present. While the symplectic potential itself may not be globally well-defined on the target space $\mathcal{M}$, the action can be recovered by Witten's construction [68], that is by integrating the closed $(d+2)$-form $(1/d!)\epsilon_{a_1 \cdots a_d} \Omega(\phi) \wedge \mathrm{d}X^{a_1} \wedge \cdots \wedge \mathrm{d}X^{a_d}$ over a spacetime with an extra dimension.

# B  Linear momentum problem in axion electrodynamics

Here we demonstrate explicitly that axion electrodynamics as defined by (52) does not possess a well-defined momentum density for any nonuniform axion background $\theta(\boldsymbol{x})$. We use the temporal gauge in order to avoid having to deal with the degeneracy of the symplectic structure

of the theory due to gauge invariance [69]. Accordingly, we treat the three components of the vector potential $\boldsymbol{A}$ as independent canonical coordinates. The full set of canonical Poisson brackets then is

$$\{A_i(\boldsymbol{x}), A_j(\boldsymbol{y})\} = \{\Pi_i(\boldsymbol{x}), \Pi_j(\boldsymbol{y})\} = 0, \qquad \{A_i(\boldsymbol{x}), \Pi_j(\boldsymbol{y})\} = \delta_{ij}\delta(\boldsymbol{x} - \boldsymbol{y}), \tag{63}$$

where $\boldsymbol{\Pi} \equiv -\boldsymbol{E} - C\theta\boldsymbol{B}$ is the vector of conjugate momentum. These imply the following Poisson brackets between the gauge-invariant field variables,

$$\{E_i(\boldsymbol{x}), E_j(\boldsymbol{y})\} = -C\epsilon_{ij}{}^k[\theta(\boldsymbol{x})\partial_k^x + \theta(\boldsymbol{y})\partial_k^y]\delta(\boldsymbol{x} - \boldsymbol{y}),$$
$$\{E_i(\boldsymbol{x}), B_j(\boldsymbol{y})\} = \epsilon_{ij}{}^k\partial_k^y\delta(\boldsymbol{x} - \boldsymbol{y}), \qquad \{B_i(\boldsymbol{x}), B_j(\boldsymbol{y})\} = 0. \tag{64}$$

The twisted Poisson bracket for the electric field is responsible for the modification of Maxwell's equations in presence of the axion background since in the temporal gauge, the Hamiltonian density itself remains unchanged by the axion coupling, $\mathcal{H} = (\boldsymbol{E}^2 + \boldsymbol{B}^2)/2$.

Next, we introduce two scaling parameters $\alpha, \beta$. The first of these, $\alpha$, will count powers of $\boldsymbol{E}$ or $\boldsymbol{B}$ in any local gauge-invariant operator. The second, $\beta$, will count the number of any additional derivatives the operator may possess. Thus, for instance, both of the modified Maxwell equations in (56) are homogeneous of order $\alpha^1\beta^1$. We note that the Poisson brackets in (64) reduce the power of $\alpha$ by 2 and increase the power of $\beta$ by 1.

Suppose now for the sake of contradiction that the theory does possess a gauge-invariant local momentum density $\boldsymbol{p}(\boldsymbol{x})$. Apart from gauge invariance itself, we demand that the corresponding integral operator $\boldsymbol{P} \equiv \int \mathrm{d}^3\boldsymbol{x}\, \boldsymbol{p}(\boldsymbol{x})$ satisfies $\{P_i, \phi(\boldsymbol{x})\} = \partial_i\phi(\boldsymbol{x})$ for any local gauge-invariant operator $\phi$, that is any local function of $\boldsymbol{E}, \boldsymbol{B}$ and a finite number of their derivatives.[6] It then follows that $\boldsymbol{p}$ must be of order $\alpha^2\beta^0$. This requires in turn that $\boldsymbol{p}$ is a quadratic function of $\boldsymbol{E}, \boldsymbol{B}$ without any derivatives. Since it should also be a vector, the only possibility is

$$\boldsymbol{p}(\boldsymbol{x}) = \zeta(\theta(\boldsymbol{x}), \boldsymbol{x})\boldsymbol{E}(\boldsymbol{x}) \times \boldsymbol{B}(\boldsymbol{x}), \tag{65}$$

where $\zeta$ is an a priori arbitrary function of $\theta$, possibly also depending explicitly on the coordinates. An explicit calculation using (64) now gives

$$\{P_i, B_j\} = \zeta\partial_i B_j + B_k(\delta_{jk}\partial_i\zeta - \delta_{ij}\partial_k\zeta). \tag{66}$$

This satisfies our requirements if and only if $\zeta = 1$, which reduces $\boldsymbol{p}$ to the usual Poynting vector. However, it is then easy to show that as a consequence of the twisted Poisson bracket for the electric field, the expected property $\{P_i, E_j\} = \partial_i E_j$ cannot be satisfied for any nonzero $C$ and nonuniform $\theta(\boldsymbol{x})$. This proves that a gauge-invariant momentum density whose integral serves as a generator of spatial translations does not exist.

## C  Coordinate-free formulation of dipole conservation laws

In this and the next appendix, we address dipole conservation laws in a coordinate-free language, first form the point of view of global symmetry and then that of background gauge invariance. This requires an appropriate geometric structure on the spacetime. We generally assume invariance under spacetime translations. For simplicity, we augment this with the

---

[6]The Poisson bracket is to be evaluated in the temporal gauge and the equality is to hold up to terms that vanish when the modified Gauss law is imposed. The reason for this provision is that in the temporal gauge, the Gauss law is no longer satisfied automatically, but rather selects a set of admissible trajectories in the phase space. Given that the Gauss law is homogeneous in $\alpha, \beta$, this subtlety does not affect the validity of our argument.

assumption of invariance under continuous spatial rotations, although this is not strictly necessary, and in most condensed-matter systems clearly is just a convenient approximation. The kind of spacetime geometry that implements these symmetries locally is sometimes referred to as *Aristotelian*. See [70,71] for some early work on coupling matter to Aristotelian spacetime background, [72–74] for more recent work in the context of fractons, and finally the recent review [75] of non-Lorentzian geometry for the bigger picture.

A $(d+1)$-dimensional manifold is said to be endowed with Aristotelian geometry if it carries a 1-form $n$ and a rank-$d$ positive-semidefinite symmetric tensor field $h$,

$$n \equiv n_\mu \mathrm{d}x^\mu, \qquad h \equiv h_{\mu\nu}\mathrm{d}x^\mu \otimes \mathrm{d}x^\nu. \tag{67}$$

Intuitively, $n$ can be thought of as defining a foliation of the spacetime, and $h_{\mu\nu}$ as giving a Riemannian metric on the spatial slices. A dual structure to that of (67) is established by a vector field $\boldsymbol{v}$ and a rank-$d$ positive-semidefinite symmetric tensor field $\tilde{h}$,

$$\boldsymbol{v} \equiv v^\mu \partial_\mu, \qquad \tilde{h} \equiv \tilde{h}^{\mu\nu}\partial_\mu \otimes \partial_\nu, \tag{68}$$

satisfying the constraints

$$v^\mu n_\mu = 1, \qquad h_{\mu\nu}v^\nu = 0, \qquad \tilde{h}^{\mu\nu}n_\nu = 0, \qquad \tilde{h}^{\mu\lambda}h_{\lambda\nu} = \delta^\mu_\nu - v^\mu n_\nu. \tag{69}$$

The structure of the Aristotelian spacetime defines an auxiliary Riemannian metric $\gamma$ via

$$\gamma \equiv h + n \otimes n, \qquad \gamma_{\mu\nu} = h_{\mu\nu} + n_\mu n_\nu. \tag{70}$$

It follows from (69) that the inverse of $\gamma_{\mu\nu}$ is $\gamma^{\mu\nu} = \tilde{h}^{\mu\nu} + v^\mu v^\nu$. The 1-form $n$ and vector $\boldsymbol{v}$, as well as the tensors $h$ and $\tilde{h}$, are then related by raising or lowering indices with $\gamma^{\mu\nu}$ and $\gamma_{\mu\nu}$.

Let us now rerun the argument of Sec. 2 using the coordinate-free language of differential geometry. We generalize the action (6) to

$$S = \int (\omega \wedge \star n - \mathscr{H}\,\mathrm{vol}), \tag{71}$$

where the Hodge dual is taken with respect to the Riemannian metric (70),[7] and $\mathrm{vol} \equiv \star 1$ is the spacetime volume form. We will assume that the 1-form $n$ satisfies

$$\mathrm{d}n = \mathrm{d}\star n = 0. \tag{72}$$

The first of these properties is a sufficient, though not necessary, condition for the Aristotelian spacetime to have a foliation in terms of spatial slices. The physical origin of the second condition will be clarified in greater detail in Appendix D. Geometrically, it ensures that the spatial slices have vanishing mean extrinsic curvature $K = \nabla_\mu v^\mu = -\mathrm{d}^\dagger n = \star\mathrm{d}\star n$, where $\mathrm{d}^\dagger$ is the codifferential. Finally, we need to introduce the notion of translation invariance in the coordinate-free language. We will assume that the spacetime is homogeneous under translations generated by a set of vector fields, $e_A^\mu$, where $A = 1, \ldots, d+1$. Our previous ansatz for the variation of the Hamiltonian part of the action, that is the second relation in (7), then generalizes to

$$\delta_\xi \int \mathscr{H}\,\mathrm{vol} \equiv \int \mathrm{d}\xi^A \wedge \star \sigma_A, \tag{73}$$

where $\xi^A$ are the components of a vector field $\boldsymbol{\xi}$ in the basis $e_A^\mu$. This defines a set of 1-forms $\sigma_A$ that can be identified with the stress tensor of the Hamiltonian $\mathscr{H}$.

---

[7]This choice is a matter of convenience. In non-Riemannian or non-Lorentzian geometries, there are multiple consistent definitions of the Hodge dual [76,77].

Next we need to evaluate the variation of the first term in (71) under the diffeomorphism $\xi$. Importantly, we are only transforming the dynamical fields $\phi^a$, not the Aristotelian spacetime background. Hence $\delta_\xi \int \omega \wedge \star n = \int (\mathcal{L}_\xi \omega) \wedge \star n$, whence

$$
\begin{aligned}
\mathcal{L}_\xi \omega \wedge \star n &\simeq -\omega \wedge (\mathcal{L}_\xi \star n) = -\omega \wedge \mathrm{d}(\iota_\xi \star n) = \omega \wedge \mathrm{d}(\iota_{n^\sharp} \star \xi^\flat) \simeq (\mathrm{d}\omega) \wedge (\iota_{n^\sharp} \star \xi^\flat) \\
&\simeq -(\iota_{n^\sharp} \mathrm{d}\omega) \wedge \star \xi^\flat = -\xi^A (\iota_{n^\sharp} \mathrm{d}\omega)_A \, \mathrm{vol} \, .
\end{aligned}
\tag{74}
$$

Here $\simeq$ denotes equality up to a total derivative. Also, we used the assumption that $\mathrm{d} \star n = 0$ and the differential-geometric identity

$$
\iota_X \star Y^\flat = -\iota_Y \star X^\flat \, ,
\tag{75}
$$

valid for any vectors fields $X, Y$. Finally, the operators $^\sharp$ and $^\flat$ indicate the so-called musical isomorphisms, raising and lowering indices respectively with $\gamma^{\mu\nu}$ and $\gamma_{\mu\nu}$. Putting all the bits together, we find that

$$
\delta_\xi S = \int \xi^A [-(\iota_{n^\sharp} \mathrm{d}\omega)_A \, \mathrm{vol} + \mathrm{d} \star \sigma_A] \, .
\tag{76}
$$

Upon using the equation of motion, we then obtain the on-shell relation

$$
\iota_{n^\sharp} \mathrm{d}\omega = e^A \star \mathrm{d} \star \sigma_A = -e^A \mathrm{d}^\dagger \sigma_A \, ,
\tag{77}
$$

where $e^A$ is the basis of 1-forms, dual to $e_A$. This is the coordinate-free generalization of (9).

It remains to combine this with the off-shell conservation of the current (10), which now reads simply $\mathrm{d}(\mathrm{d}\omega) = 0$. Here we use the projection–rejection decomposition of $\mathrm{d}\omega$,

$$
\mathrm{d}\omega = \iota_{n^\sharp}(n \wedge \mathrm{d}\omega) + n \wedge (\iota_{n^\sharp} \mathrm{d}\omega) \, .
\tag{78}
$$

Taking the exterior derivative and using the assumption $\mathrm{d}n = 0$ along with (77) leads to our final result for the dipole-type conservation law in the coordinate-free language,

$$
\boxed{\mathcal{L}_{n^\sharp}(n \wedge \mathrm{d}\omega) + n \wedge \mathrm{d}(e^A \mathrm{d}^\dagger \sigma_A) = 0 \, .}
\tag{79}
$$

In a flat Aristotelian spacetime, this reduces to our previous results (1) and (15). The advantage of the form (79) is that it is manifestly independent of the number of spatial dimensions. Moreover, it generalizes the global dipole conservation law (15) in a flat Aristotelian spacetime to any Aristotelian spacetime subject to the conditions (72) and (73). The condition (72) is manifestly satisfied for instance for spacetimes of the type $\mathbb{R} \times M$ where $M$ is a $d$-dimensional spatial manifold, provided we set $n = \mathrm{d}t$, where $t$ is the global time variable. In this case, $\star n$ is just the volume form on $M$. Perhaps even more interestingly, our derivation may be readily further generalized to manifolds that do not possess any translation invariance whatsoever. All we have to do is to replace (73) with $\delta_\xi \int \mathcal{H} \, \mathrm{vol} = \int \xi^A \wedge \star \tau_A$, where $e_A^\mu$ is now an arbitrary local frame and $\tau_A$ a set of functions defining the variation of the Hamiltonian along $e_A^\mu$. The only change to (79) is that $\mathrm{d}^\dagger \sigma_A$ has to be replaced with $\tau_A$. The resulting local differential law does not lead to conservation of multipole moments of the topological charge (2). It may however be useful for constraining the motion of topological solitons on curved surfaces, which has been studied in the mathematical physics literature, see for instance [78–80].

Let us conclude with the remark that the generalization of multipole conservation laws to curved spacetime backgrounds has recently been investigated in [73, 74]. It is therefore worthwhile to clarify the difference of the setups used therein and here. Namely, in the literature, fracton theories typically come, either explicitly or implicitly, equipped with complex

fields, carrying the scalar charge $Q$. Moreover, it is most common to treat the dipole moment of this charge separately from the operator of spatial momentum. On the contrary, here the two vector opeartors are identified via (4). In addition, the charge $Q$ is topological and therefore does not act on the elementary fields $\phi^a$. Finally, the general class of conservation laws of the type (15) does not seem to have been studied elsewhere at all. Another, more operational difference is that in the high-energy physics literature, symmetries are often studied via background gauging, whereas we have so far used solely the physical, global translation invariance. We shall elaborate on the role of background gauge invariance of the action in the context of the present paper in the following appendix.

## D  Gauging dipole symmetry: volume-preserving diffeomorphisms

Suppose we would like to encode the global symmetries of an EFT in terms of background gauge invariance of its generating functional. Putting aside temporal translations, the global symmetries of interest to us are all displayed in Fig. 1.[8] All the generators are various moments of the topological charge $Q$. It therefore appears to be possible to gauge all the symmetries simultaneously by considering the action of the generic moment $Q_\lambda$ defined by (18). But according to (21), the transformation of the canonical fields $\phi^a$ generated by $Q_\lambda$ corresponds to a translation $x^i \rightarrow x^i + \xi^i(\boldsymbol{x}, t)$ with $\xi^i = -\epsilon^{ij}\partial_j\lambda$. This is an example of a volume-preserving diffeomorphism (VPD), since $\boldsymbol{\nabla} \cdot \boldsymbol{\xi} = 0$. We therefore expect that gauging the dipole symmetry will lead to invariance of the classical action of the theory under spatial VPDs [15].

Let us now formalize the above intuitive argument. The topological current $J = \star\Omega = \star\,\mathrm{d}\omega$, generalizing (10), can be coupled to a $(d-1)$-form background gauge field, $A$. This extends the action (71) to

$$S = \int (\omega \wedge \star n + A \wedge \star J - \mathscr{H}\,\mathrm{vol}) = \int (\omega \wedge \star n + A \wedge \mathrm{d}\omega - \mathscr{H}\,\mathrm{vol})\,. \tag{80}$$

The Hamiltonian density $\mathscr{H}$ contains by construction only spatial derivatives of $\phi^a$. The Hamiltonian term in the action can accordingly be made invariant under spatial diffeomorphisms in the usual way by coupling in the spatial metrics $h_{\mu\nu}$ and $\tilde{h}^{\mu\nu}$. We shall therefore focus on the first two terms in (80).

The term $A \wedge \mathrm{d}\omega$ is obviously invariant under the gauge transformation $\delta A = \mathrm{d}\lambda$, where $\lambda$ is a $(d-2)$-form gauge parameter. At the same time, the term is also manifestly diffeomorphism-invariant. The same is however not true for the $\omega \wedge \star n$ term, as long as we wish to keep the 1-form background $n$ fixed. As pointed out in [15], the invariance under spatial VPDs can be recovered thanks to cancellation between the transformations of $\omega \wedge \star n$ and $A \wedge \mathrm{d}\omega$. With this in mind, we consider the following ansatz,

$$\delta\omega = \mathcal{L}_\xi\omega\,, \qquad \delta A = \mathcal{L}_\xi A + \Lambda\,, \tag{81}$$

where $\xi \equiv \xi^\mu\partial_\mu$ is an as yet unspecified vector field, and $\Lambda$ an as yet unspecified $(d-1)$-form. Assuming, as mentioned above, the invariance of the Hamiltonian term, the variation of the action becomes

$$\delta_\xi S = \int \omega \wedge [\mathrm{d}\Lambda - \mathcal{L}_\xi(\star n)]\,. \tag{82}$$

At this point, we need the assumption $\mathrm{d}\star n = 0$, which we made rather casually in Appendix C. This is, in fact, essential in case $\Omega = \mathrm{d}\omega$ is cohomologically nontrivial, and thus $\omega$ is not

---

[8]The figure was designed with the special case of $d = 2$ spatial dimensions in mind. However, the analysis below applies without change to any $d$.

globally well-defined on the target space $\mathcal{M}$. Namely, it guarantees that the $\omega \wedge \star n$ term in the action can be recovered by integration of the closed form $\Omega \wedge \star n$ over a spacetime with one extra dimension à la Witten [68]. The fact that for cohomologically nontrivial $\Omega$ the EFT cannot be coupled to an arbitrary Aristotelian background reflects the LMP, as explained in Sec. 7.

With the assumption $d \star n = 0$, we now use the Cartan magic formula along with (75) to rewrite (82) as

$$\delta_\xi S = \int \omega \wedge d(\Lambda + \iota_{n^\sharp} \star \xi^\flat).$$ (83)

It might appear that we can choose any $\xi$ and then simply set $\Lambda = -\iota_{n^\sharp} \star \xi^\flat$ to keep the action unchanged. But this may spoil the assumed invariance of the Hamiltonian term. A suitable coordinate-free formulation of the expected VPDs is

$$\xi^\flat = \star(n \wedge d\lambda), \qquad \Lambda = (-1)^d n \wedge (\iota_{n^\sharp} d\lambda),$$ (84)

where $\lambda$ is again an arbitrary $(d-2)$-form gauge parameter. By applying the projection–rejection decomposition (78) to $d\lambda$ instead of $d\omega$, we readily find that $\Lambda + \iota_{n^\sharp} \star \xi^\flat = (-1)^d d\lambda$, which guarantees the invariance of the action. Note that we have not used anywhere the assumption that $dn = 0$. This is only needed to ensure that the diffeomorphism generated by the vector field $\xi$ actually is a VPD, thanks to $\nabla \cdot \xi = -d^\dagger \xi^\flat = \star d \star \xi^\flat = (-1)^d \star (dn \wedge d\lambda)$.

Let us unpack the meaning of the generalized VPD (84) in a simple, familiar setting. We restrict to a trivial Aristotelian background in $d = 2$ spatial dimensions. Here $n = dt$ and the action (80) becomes

$$S = \int d^2\boldsymbol{x} dt \left( \omega_0 + A_\mu J^\mu - \mathcal{H} \right),$$ (85)

where $J^\mu$ is defined by (10). This extends our original action (6) by adding a source term for the topological current. The transformation parameters (84) reduce to

$$\xi^i = -\epsilon^{ij} \partial_j \lambda, \qquad \Lambda = \partial_0 \lambda dt.$$ (86)

The corresponding transformations (81) translate to

$$\delta \omega_0 = \xi^i \partial_i \omega_0 + \omega_i \partial_0 \xi^i, \qquad \delta A_0 = \xi^i \partial_i A_0 + A_i \partial_0 \xi^i + \partial_0 \lambda, \qquad \delta A_i = \xi^j \partial_j A_i + A_j \partial_i \xi^j.$$ (87)

Up to an overall rescaling of $\xi^i$ and $\lambda$, this agrees with the "nonlinear higher-rank symmetry" put forward in [15]. The symmetry corresponds to a simultaneous spatial VPD and a gauge transformation of the temporal component of the background gauge field $A_\mu$.

This completes the argument demonstrating how the global dipole-type symmetry analyzed in this paper can be embedded in a background-gauge-invariant setting. For the sake of completeness, let us add that that the background gauge transformations (84) can be used to obtain a Ward–Takahashi identity for the generating functional of the theory, which in turn recovers the local conservation law (15). See [15] for further details.

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
