# Peer review of "Dipole symmetries from the topology of the phase space and the constraints on the low-energy spectrum"

_SciPost Physics, doi:SciPost Phys. 16, 051 (2024)_

## Round 1 · Referee Report · Anonymous · 2023-5-21

Report
My feeling after reading the paper is that there is some interesting theory idea being developed, but I am rather concerned that some of the punchlines of the paper may have been misinterpreted. The presentation can, in places, also be improved. I think the authors should address the points below before the paper be considered for publication.
Starting in Section 2, the notation is somewhat confusing: the symplectic form is presumably actually an infinite-dimensional object since the *fields* themselves are the degrees of freedom on phase space, but the authors seem to treat \omega_a as a 1-form on a finite-dimensional manifold. I suppose this is allowed in the presence of somewhat stronger assumptions, i.e. L(phi) has no terms of the form (d_t phi)^2 (d_x phi)^2. Later on, the authors seem to implicitly assume this in Eq. (17). I think a clearer description of what Lagrangians are allowed and developing the Hamiltonian formalism would make the first part of the paper easier to follow.
I would also remark, on a related note, that the authors don't a priori rely on not having any (d_t phi)^4 in L, they just need a notion of Legendre transform to exist.
The algebra of the Q_{lambda}s is infinite-dimensional. I see that some simple subset of it reproduces the dipole + trace quadrupole part of the multipole algebra. But why is this the only interesting part? Perhaps the other parts do not commute with H[phi]? If so, this point needs to be emphasized more in Section 3.
A related point is that if I have to put the theory on T^2, the operator L no longer makes sense (I don't have continuous rotational symmetry on a torus, just a discrete rotational symmetry at best). If I put the theory on S^2, I probably shouldn't define translations P_i either. These points should be included below Eq. (13), since they are big obstructions to one of the ways that the authors might try to make this topological charge Q non-trivial.
I really like the idea behind the examples in Section 4.2-4.3, which I think make the construction feel more physical. However, my understanding is that if you want to study a superfluid (really, any kind of) vortices on T2, you must have total (signed) vortex number equal to 0. One way to think of this is to try and place a single vortex: then draw a small loop C around the vortex. Evaluate \int_C dx . \nabla \theta, first by counting the number of vortices inside of C, and then by counting the number outside C (i.e. on the rest of the torus, which is also the "inside" of the loop since the manifold is closed). These numbers do not agree unless the total vortex number on T^2 vanishes: i.e. there is a vortex somewhere else. But of course to the authors, this seems like restricting to the Q=0 sector, which means that there is no problem having well-defined global translations after all! And, according to the authors in Section 3 below Eq. (13), Q is only interesting on manifolds with a non-trivial 2-cycle, so their construction can't be salvaged by going to R^2 as far as I can tell. If the authors must fix non-trivial boundary conditions at infinity on R^2, then the fact that translation invariance must be broken seems less surprising to me (see next paragraph).
The discussion of the linear momentum problem in the paper seems pretty abstract and there are not that many concrete calculations in the main text. At a broader level, my feeling is that it makes more sense to say that in some theories, we are just interested in modifying the algebra of momenta (e.g. in a low energy EFT for vortices where vortex coordinates x and y become conjugate). But putting aside the jargon, I appreciate the authors' perspective and think it seems interesting. If one way that the authors' result can be understood in a less formal way is simply to note that the existence of a skyrmion field configuration in the plane (e.g. in the ferromagnet) must necessarily break translations by placing the skyrmion somewhere in space (and trying to move the skyrmion, i.e. by acting with a translation, would change the values of the conserved momenta), then I think the authors should give this more physical picture for their result. This last point could be related to recent arguments about spontaneous dipole symmetry breaking at finite density: see e.g. arXiv:2205.01132, 2301.02680; and boosts being spontaneously broken in more general fluids (see references therein).
Author: Tomas Brauner on 2023-07-01 [id 3776]
(in reply to Report 1 on 2023-05-21)
We would like to thank the referee for reading carefully the manuscript and for providing feedback that helps us improve the text. Below, we address one by one the points raised by the referee. Wherever appropriate, we briefly describe the intended modifications of the manuscript, taking into account the referee's feedback.
Starting in Section 2, the notation is somewhat confusing: the symplectic form is presumably actually an infinite-dimensional object since the fields themselves are the degrees of freedom on phase space, but the authors seem to treat $\omega_a$ as a 1-form on a finite-dimensional manifold. I suppose this is allowed in the presence of somewhat stronger assumptions, i.e. $\mathscr{L}$ has no terms of the form $(\partial_t\phi)^2(\partial_x\phi)^2$. Later on, the authors seem to implicitly assume this in Eq. (17). I think a clearer description of what Lagrangians are allowed and developing the Hamiltonian formalism would make the first part of the paper easier to follow.
Reply: Here the referee points out two separate issues, namely the assumptions we make on the dependence of the Lagrangian on field derivatives, and the notation for the symplectic formalism we use. The answer to these points is however related. First, the precise condition under which our analysis holds is that the Lagrangian density does not contain any higher time derivatives or mixed temporal-spatial derivatives. It is therefore assumed to be a local function of the fields, their first time derivatives, and arbitrarily high spatial derivatives. This makes it possible to perform the Legendre transform from generalized velocities to generalized momenta locally and algebraically. In order to make this point clear, we will expand the introductory part of Sec. 2 in a way that makes the transition from Lagrangian to Hamiltonian formalism explicit.
Under the above assumption, the term in the Hamiltonian form of the action, linear in time derivatives, is strictly local in that the coefficient of $\partial_0\phi^a$ is a mere function on the target space $\mathcal{M}$. Treating $\omega_a$ as a 1-form on $\mathcal{M}$ is therefore correct and consistent. The only issue here might be that calling $\omega_a$ the symplectic potential and $d\omega_a$ the symplectic 2-form might be an abuse of terminology. As the referee points out, these objects generally act on the infinite-dimensional phase space of the field theory. We will add a footnote on page 4 of the manuscript that alerts the reader of this possible ambiguity in the terminology.
The algebra of the $Q_{\lambda}$s is infinite-dimensional. I see that some simple subset of it reproduces the dipole + trace quadrupole part of the multipole algebra. But why is this the only interesting part? Perhaps the other parts do not commute with $H$? If so, this point needs to be emphasized more in Section 3.
Reply: As we point out in Appendix D, $Q_\lambda$ generates a spatial volume-preserving diffeomorphism $x^i\to x^i+\xi^i(x)$ with $\xi^i=-\epsilon^{ij}\partial_j\lambda$. In this sense, all the $Q_\lambda$s are interesting, as they enter the background-gauge formulation of theories with global dipole symmetry. Whether or not a given $Q_\lambda$ Poisson-commutes with the Hamiltonian depends on the spatial symmetries of the latter. We do not mention this in Sec. 3 because it is irrelevant for the discussion therein.
A related point is that if I have to put the theory on $T^2$, the operator $L$ no longer makes sense (I don't have continuous rotational symmetry on a torus, just a discrete rotational symmetry at best). If I put the theory on $S^2$, I probably shouldn't define translations $P_i$ either. These points should be included below Eq. (13), since they are big obstructions to one of the ways that the authors might try to make this topological charge $Q$ non-trivial.
Reply: The class of field theories we consider is defined on the Euclidean space $\mathbb{R}^d$ without any particular boundary condition. The discussed boundary conditions that effectively compactify $\mathbb{R}^d$ are not an a priori part of the definition of the theory; they are merely used to select potentially interesting field configurations. We will improve the discussion between Eqs. (13) and (14) in a way that emphasizes the distinction between imposing a boundary condition on specific field configurations and on the entire theory. We thank the referee for pointing out that this distinction was not made clearly enough in the original manuscript.
I really like the idea behind the examples in Section 4.2-4.3, which I think make the construction feel more physical. However, my understanding is that if you want to study a superfluid (really, any kind of) vortices on $T^2$, you must have total (signed) vortex number equal to $0$. One way to think of this is to try and place a single vortex: then draw a small loop $C$ around the vortex. Evaluate $\int_Cd\vec x\cdot\vec\nabla\theta$, first by counting the number of vortices inside of $C$, and then by counting the number outside $C$ (i.e. on the rest of the torus, which is also the ``inside'' of the loop since the manifold is closed). These numbers do not agree unless the total vortex number on $T^2$ vanishes: i.e. there is a vortex somewhere else. But of course to the authors, this seems like restricting to the $Q=0$ sector, which means that there is no problem having well-defined global translations after all! And, according to the authors in Section 3 below Eq. (13), $Q$ is only interesting on manifolds with a non-trivial 2-cycle, so their construction can't be salvaged by going to $\mathbb{R}^2$ as far as I can tell. If the authors must fix non-trivial boundary conditions at infinity on $\mathbb{R}^2$, then the fact that translation invariance must be broken seems less surprising to me (see next paragraph).
Reply: As we explained above, the boundary conditions are used merely as a tool to identify interesting field configurations. Continuous translations and rotations can therefore be defined in the usual way without having to worry about boundary conditions. Likewise, there is no topological constraint on the total winding number of vortices in a superfluid.
The discussion of the linear momentum problem in the paper seems pretty abstract and there are not that many concrete calculations in the main text. At a broader level, my feeling is that it makes more sense to say that in some theories, we are just interested in modifying the algebra of momenta (e.g. in a low energy EFT for vortices where vortex coordinates $x$ and $y$ become conjugate). But putting aside the jargon, I appreciate the authors' perspective and think it seems interesting. If one way that the authors' result can be understood in a less formal way is simply to note that the existence of a skyrmion field configuration in the plane (e.g. in the ferromagnet) must necessarily break translations by placing the skyrmion somewhere in space (and trying to move the skyrmion, i.e. by acting with a translation, would change the values of the conserved momenta), then I think the authors should give this more physical picture for their result. This last point could be related to recent arguments about spontaneous dipole symmetry breaking at finite density: see e.g. arXiv:2205.01132, 2301.02680; and boosts being spontaneously broken in more general fluids (see references therein).
Reply: While the first part of the paper largely revolves around the central extension of the momentum algebra, in Secs. 6 and 7 this is no longer the main target but serves merely as a tool to approach the linear momentum problem (LMP). The gist of the LMP is not the central extension of the momentum algebra per se. For instance, in superfluids, the presence of a vortex makes the components of momentum noncommutative yet there is no LMP. Likewise, the problem is not that somehow global translations are not well-defined anymore. (They are still well-defined.) Finally, LMP is not about spontaneous breaking of translation invariance. Even when translations are spontaneously broken, one can still have an exact (classical) local momentum conservation law. Rather, the presence of LMP indicates that there is no local momentum density that would be well-defined and nonsingular for all smooth field configurations in the phase space of the theory, and hence no generally valid local momentum conservation law.
We would like to stress that the LMP is an old, long-standing problem in condensed matter systems, as described in Sec. 7.2 in detail. The concrete and physical examples of the LMP are also discussed there and in the given references. The main purpose of our discussion of the LMP is rather to provide a generic theorem, independent of microscopic details, for the presence of additional light modes in the system whenever the short-distance physics is governed by a translationally invariant local field theory. We also provide a new example of the LMP for higher-form symmetry with concrete calculations in Sec. 8. We thus believe that the true nature of the LMP and its implications are discussed in the manuscript rather carefully. However, in order to help the reader appreciate the long history behind the LMP, we will add some relevant references early in Sec. 6 where the LMP is introduced.
Author: Tomas Brauner on 2023-10-31 [id 4087]
(in reply to Report 2 on 2023-08-30)We would like to thank the referee for the positive overall evaluation of our manuscript, and for very interesting suggestions how to further extend its scope. Below, we comment on all the five points raised by the referee, outlining the corresponding changes in the text. For the referee's convenience, we attach the revised version of the manuscript where all the changed parts of the text are highlighted in red.
Reply: We thank the referee for pointing out that this point might lead to confusion. In order to avoid misunderstanding, we have appended at the end of Sec. 2 a warning to the reader that (15) should not be interpreted as a multipole-type conservation law.
Reply: Following the suggestion by the referee, we have expanded the discussion in the last paragraph of Appendix C. The text, now also including the suggested reference 2304.09596, newly stresses that none of the cited works dealing with fracton systems on curved spacetimes seems to cover the situation addressed in our paper, including the generalized conservation law (15). Namely, in the mentioned references, there are always elementary excitations carrying the scalar charge $Q$. Also, the fracton algebra considered includes separate operators of momentum and dipole moment. In our case, the charge $Q$ is topological and only affects extended field configurations such as solitons. Moreover, the operators of dipole moment and momentum are identified via our (4). As a side remark, we do mention in Appendix D, below (82), an issue with coupling our effective theory to curved spacetime backgrounds. This is closely related to the linear momentum problem, which is one of the central themes of our paper.
A separate question raised by the referee is how to obtain our dipole-type conservation law from a gauging procedure. This question was addressed previously by Du et al. (our Ref. [15]). They showed how, in two and three spatial dimensions, invariance of the generating functional of a theory under volume-preserving spatial diffeomorphisms leads to a Ward-Takahashi identity that in turn recovers the dipole conservation law. Since this question was already thoroughly addressed by Ref. [15], we limited the discussion in Appendix D to a demonstration how the invariance under volume-preserving diffeomorphisms naturally arises from the global symmetry framework, in which the rest of our paper is phrased. However, for the sake of completeness, we have newly added to Appendix D a comment pointing out that the background gauge formalism can also be used to extract the dipole conservation law.
Reply: First of all, the conservation of winding number of vortices in ordinary superfluids may itself be a higher-form symmetry depending on the number of spacetime dimensions. This case is already covered by our conservation law (15) and the discussion in Sec. 4.3. We have not considered generalized superfluids where the dynamical Nambu-Goldstone field is a higher-form field, with the single exception of the discussion of axion electrodynamics in Sec. 8. While we agree with the referee that this would be an interesting further extension of our work, we cannot make any more specific comments at this stage.
Reply: In this comment, the referee raises several different points. The first of these is the validity of our conclusions when the Lagrangian receives corrections, possibly of higher order in derivatives. This issue is already to some extent addressed in the conclusions (Sec. 9), where we discuss a specific type of perturbation that is of second order in time derivatives. We have newly extended the discussion in Sec. 9 to clarify that the existence of a local dipole-type conservation law really relies just on translation invariance. Other than that, the only assumption needed is that the charge density and current in (1) or (15) are well-defined on the target space of the theory. This latter assumption may be violated for instance in theories with a gauge redundancy, as we demonstrate in Sec. 8. The additional assumption of absence of higher-order time derivatives is only needed from Sec. 3 on when we analyze the algebra of conserved charges using the symplectic formalism. Altogether, we therefore make a robust prediction of the existence of an exact local dipole-type conservation law, although the precise expressions for the corresponding charge density and current may be modified by perturbations.
The above said, though, there are perturbations that may lead to explicit breaking of our dipole-type conservation law, essentially by making the integral charge $Q$ either unphysical or not conserved. This however requires extending the phase space of the theory by new dynamical degrees of freedom, rather than merely adding some new operators to the Lagrangian. In the revised version of the manuscript, we have added a long paragraph to Sec. 9 that explains this mechanism, including the superfluid as a concrete example.
The second point raised by the referee is whether our dipole-type conservation law survives in systems out of equilibrium with effects of dissipation taken into account. Since we work exclusively within ordinary Lagrangian field theory, we unfortunately cannot make any definitive statement in this regard. We expect that dissipation, which amounts to higher-order derivative corrections of the stress tensor and current, would not spoil the dipole conservation law, since it is a consequence of the translational invariance of the theory, under the condition that the charge density and current are well-defined on the target space of the theory, as discussed in Sec. 9. To demonstrate the dipole conservation law more rigorously within the action principle, one would need to use the effective field theory for dissipative fluids (following, for instance, 1511.03646), but this is beyond the scope of the present paper.
Finally, the referee wonders whether the dipole-type symmetry in our paper may be generally useful as a guiding principle for construction of Lagrangians. Here we would again point out that the only symmetries we assume are spatial translations and rotations, and the symmetry associated with the scalar charge $Q$. The latter is however topological and thus does not generate any transformations of the elementary fields. (As we point out in Sec. 3, all Poisson brackets of $Q$ vanish.) This kind of symmetry indeed cannot be used as a "guiding principle," but rather should be treated as emergent.
Reply: We have added the reference as suggested by the referee.
Attachment:
manuscript_changes.pdf

---

## Round 1 · Referee Report · Anonymous · 2023-8-30

Report
The paper provides an interesting EFT perspective on why dipole conservation laws have been reported to appear in various condensed matter systems. I recommend it for publication but think that the authors could address a few points before:
1 - The case d=2 does indeed lead to the standard dipole conservation law that has been discussed in many earlier papers. However the case of d>2 with the conservation law (15) is somewhat different because (d-2) indices are form indices and hence anti-symmetric under the exchange of any two indices. I do notice that the authors refer to this as a "dipole-type conservation law" but perhaps it would be useful to elucidate that this is not arising from the multipole algebra.
2 - In relation to the conservation law (15), the authors mention the coupling to curved backgrounds and have an appendix C with more details on this coupling. I think it should be stressed that none of the papers that the authors refer to have addressed such dipole conservation (15) precisely because of the form indices. How to couple these theories to curved space in d>2 is therefore not really addressed as far as I can tell. The authors may want to comment on this. In particular it is known that there are issues with coupling to curved space when considering the fracton algebra as in references [66,67] and one may wonder whether generating (15) will come with similar (or more) issues. In addition, the authors may want to consider citing the recent paper arXiv:2304.09596 where the gauging of the fracton algebra and coupling to curved space in generality was considered. What algebra has to be gauged to generate (15)?
3 - The authors discuss also applications to higher-form symmetries. Have the authors thought about whether the conservation law (15) applies to higher-form vorticies? These were recently discussed in detail in arXiv:2301.09628. It would be interesting to look at such cases since it would give additional relevant to (15) if that turns out to be the relevant conservation law in these situations.
4 - The authors make a specific assumption of their Lagrangian, in particular, Lagrangians only have one time derivative. The question that, I believe, still remains in the literature is whether such dipole symmetry, say for d=2 in superfluid EFT, is exact or just a consequence of a specific truncation of the Lagrangian. For instance, it is expected that in superfluid EFT once going off equilibrium and looking at dissipative effects, additional terms with higher time derivatives appear. Will these not spoil the dipole symmetry? What I am trying to say is that perhaps the authors should be more careful with certain statements as their analysis, as far as I can tell, will only work as an approximation to such systems, including superfluid EFT for which dipole symmetry may not be the actual symmetry of the system and hence may not work as a guiding principle for building the Lagrangian at increasingly higher orders in gradients.
5 - I think that the paper arXiv:1802.09512 should be cited along with reference [41].

---

## Round 2 · Referee Report · Anonymous (Referee 2) · 2023-11-2

Report

The authors have successfully addressed all my points. I recommend the paper for publication.
  • validity: -
  • significance: -
  • originality: -
  • clarity: -
  • formatting: -
  • grammar: -

Author:  Tomas Brauner  on 2024-01-25  [id 4282]

(in reply to Report 1 on 2023-11-02)

We are delighted that the referee recommends publication of our paper, and thank the referee for having invested their time in reading the revised manuscript.

---

## Round 2 · Referee Report · Anonymous (Referee 1) · 2024-1-4

Report

Given the previous report and the authors' careful reply, I am fine with publication of the paper in SciPost Physics.

If the authors wish, I would suggest commenting a bit more on two points.

The first is on the discussion in Section 9, around Equation (59-60). My intuition is that somehow this dipole charge must end up trivial, because ordinarily one expects that having dipole symmetry leads to some important constraints on EFTs (changes to exponents in dynamics, Mermin-Wagner, etc...), but if I give you a generic field theory with (d_0 phi)^2 - V(phi), I don't expect such constraints to be relevant. I think the authors should remark on this -- some comments are made at the end of the paragraph but I found them a bit hard to interpret. To be direct here: is there anything useful that this "hidden" dipole symmetry tells me about the "ordinary" theory I suggested above.

Second and relatedly, I don't think it's accidental at all that the interesting examples described by the authors all have Lagrangians of the form Eq. (6). It seems that then the appearance of a multipole algebra is quite similar to Ref. [6], although I think the higher-dimensional generalization in this paper looks new. Anyway, the fact that the authors' model has P_i playing the same role as D_i, and thus obtaining a multipole algebra, seems to me closely related to the Lagrangian being of the form Eq. (6). For example, if I wrote my generic simple theory from the previous paragraph in the form of Eq. (6) I would need to introduce an additional degree of freedom beyond phi, and that might qualitatively change how I would interpret things such as the LMP?
  • validity: good
  • significance: good
  • originality: high
  • clarity: high
  • formatting: excellent
  • grammar: excellent

Author:  Tomas Brauner  on 2024-01-25  [id 4283]

(in reply to Report 2 on 2024-01-04)

We would like to thank the referee for having read the revised version of the manuscript and recommended it for publication. Below, we address the two points that the referee draws attention to.

The first is on the discussion in Section 9, around Equation (59-60). My intuition is that somehow this dipole charge must end up trivial, because ordinarily one expects that having dipole symmetry leads to some important constraints on EFTs (changes to exponents in dynamics, Mermin-Wagner, etc...), but if I give you a generic field theory with $(\partial_0\phi)^2-V(\phi)$, I don't expect such constraints to be relevant. I think the authors should remark on this - some comments are made at the end of the paragraph but I found them a bit hard to interpret. To be direct here: is there anything useful that this "hidden" dipole symmetry tells me about the "ordinary" theory I suggested above.

Reply: The part of Sec. 9 that the referee points to revolves around the relation between local momentum conservation and the dipole conservation law in presence of the linear momentum problem (LMP). This issue is intimately related to the global structure of the target space $\mathcal M$ from which the canonical variables of the theory take values. Namely, as we show, a sufficient condition for the appearance of LMP is that the symplectic form of the theory (treated as a 2-form on $\mathcal M$) is cohomologically nontrivial. As we mention at the beginning of Sec. 2 (see also the example of antiferromagnets, discussed at the end of Sec. 4.1), for theories that have a well-defined, second-order Lagrangian description with a manifold $\mathcal N$ as the target space, the target space $\mathcal M$ of the Hamiltonian formulation is the cotangent bundle $T^*\mathcal N$. The corresponding symplectic form is exact, leading to the absence of LMP. For the type of "generic field theory" suggested by the referee, the local momentum conservation is therefore problem-free. In such cases, the dipole conservation law is literally just the curl of the local momentum conservation law, the scalar charge density being nothing but a generalized vorticity. There may still be field configurations satisfying a nontrivial boundary condition at spatial infinity for which the integral charge $Q$ is nonzero, and the momentum algebra is thus centrally extended. This however also comes with some topological constraints, similarly to the superfluids discussed in Sec. 4.3, where the presence of vortices is associated with a nontrivial first cohomology group of the vacuum manifold. Finally, there is a class of theories that do not feature LMP and where the scalar charge $Q$, if well-defined, always vanishes. This is the "default" option, likely realized for all theories with a single real-valued scalar field. For such theories, the dipole symmetry, both locally and globally, does not carry any new information beyond what follows from translation invariance alone. In the newly revised version of the manuscript, we make the distinction between these different classes of theories more explicit by expanding the discussion in the paragraph below Eq. (60).

Second and relatedly, I don't think it's accidental at all that the interesting examples described by the authors all have Lagrangians of the form Eq. (6). It seems that then the appearance of a multipole algebra is quite similar to Ref. [6], although I think the higher-dimensional generalization in this paper looks new. Anyway, the fact that the authors' model has $P_i$ playing the same role as $D_i$, and thus obtaining a multipole algebra, seems to me closely related to the Lagrangian being of the form Eq. (6). For example, if I wrote my generic simple theory from the previous paragraph in the form of Eq. (6) I would need to introduce an additional degree of freedom beyond $\phi$, and that might qualitatively change how I would interpret things such as the LMP?

Reply: In Ref. [6], a generalized vorticity conservation equivalent to the dipole conservation law is derived by taking the curl of momentum conservation. However, for us - as stressed in the conclusions - this is just a "mnemonic" since much of our paper focuses on theories with LMP where the local momentum density is ill-defined. What is new in our paper is the direct and explicit connection between the dipole conservation law, the central extension of momentum algebra, and the symplectic structure of the theory; this appeared in Ref. [6] only through specific examples and not as a universal feature of the Hamiltonian description of field theory.

It is indeed not accidental that the interesting examples in our paper have a first-order Lagrangian formulation of the form (6). This is because, as pointed out in our response to the referee's previous point, theories that have a nonsingular second-order Lagrangian formulation tend to have an exact symplectic form. Moreover, as the referee correctly states, writing such theories in the form (6) requires adding new degrees of freedom, namely the generalized momenta. This is discussed on the concrete example of antiferromagnets in Sec. 4.1.

Altogether, it seems to us that this point of the referee raises two separate issues. The first of these is the scope of the part of the field theory landscape for which our analysis has nontrivial consequences. We believe this is already addressed sufficiently by our response to the referee's first point. The second issue is the novelty of our results. In order to address this, we have expanded the last paragraph of Sec. 5 where we briefly summarize the results of the first half of the paper. In the revised text, we stress the universality of our results, whereby the local dipole conservation law only relies on translation invariance, and the central extension of the algebra of spatial translations in addition on the symplectic structure of the theory.

Attachment:

manuscript_changes.pdf

---

## Round 2 · List of Changes

A version of the revised manuscript where all the changed parts of the text are highlighted in color is attached to the reply to the second referee report.

---

## Round 3 · Author Response

We have addressed the point raised by the second referee in their report, and hope that the revised manuscript can be accepted for publication.

Sincerely,
The Authors

---

## Round 3 · List of Changes

We have expanded the last paragraph of Sec. 5 to stress the novelty of our results concerning dipole symmetries. In addition, we have rewritten the paragraph below Eq. (60). Other than that, we have only updated references [4] and [11] with publication data.

---

## Editorial Decision

published